# Genetic switches designed for eukaryotic cells and controlled by serine integrases

Mayna S. Gomide [1,2,3], Thais T. Sales[1,2], Luciana R. C. Barros [4], Cintia G. Limia [4], Marco A. de Oliveira[1,2], Lilian H. Florentino[1], Leila M. G. Barros [1], Maria L. Robledo[4], Gustavo P. C. José[1], Mariana S. M. Almeida[1], Rayane N. Lima[1], Stevens K. Rehen [5,6], Cristiano Lacorte[1], Eduardo O. Melo[1,7], André M. Murad[1], Martín H. Bonamino [4,8✉], Cintia M. Coelho [9✉] & Elibio Rech [1✉]

Recently, new serine integrases have been identified, increasing the possibility of scaling up genomic modulation tools. Here, we describe the use of unidirectional genetic switches to evaluate the functionality of six serine integrases in different eukaryotic systems: the HEK 293T cell lineage, bovine fibroblasts and plant protoplasts. Moreover, integrase activity was also tested in human cell types of therapeutic interest: peripheral blood mononuclear cells (PBMCs), neural stem cells (NSCs) and undifferentiated embryonic stem (ES) cells. The switches were composed of plasmids designed to flip two different genetic parts driven by serine integrases. Cell-based assays were evaluated by measurement of EGFP fluorescence and by molecular analysis of *attL/attR* sites formation after integrase functionality. Our results demonstrate that all the integrases were capable of inverting the targeted DNA sequences, exhibiting distinct performances based on the cell type or the switchable genetic sequence. These results should support the development of tunable genetic circuits to regulate eukaryotic gene expression.

[1] Brazilian Agriculture Research Corporation – Embrapa – Genetic Resources and Biotechnology – CENARGEN, Brasília 70770917 DF, Brazil. [2] Department of Cell Biology, Institute of Biological Science, University of Brasília, Brasília 70910900 DF, Brazil. [3] School of Medicine, Federal University of Juiz de Fora, Juiz de Fora 36036900 MG, Brazil. [4] Molecular Carcinogenesis Program, Research Coordination, National Cancer Institute (INCA), Rio de Janeiro 20231050 RJ, Brazil. [5] D'Or Institute for Research and Education (IDOR), Rio de Janeiro 22281100 RJ, Brazil. [6] Institute of Biomedical Sciences, Federal University of Rio de Janeiro, Rio de Janeiro 21941902 RJ, Brazil. [7] Graduation Program in Biotechnology, Federal University of Tocantins, Gurupi 77402970 TO, Brazil. [8] Vice-Presidency of Research and Biological Collections (VPPCB), FIOCRUZ – Oswaldo Cruz Foundation Institute, Rio de Janeiro 21040900 RJ, Brazil. [9] Department of Genetic and Morphology, Institute of Biological Science, University of Brasília, Brasília 70910900 DF, Brazil. ✉email: mbonamino@inca.gov.br; cintiacoelhom@unb.br; elibio.rech@embrapa.br

Recombinant DNA technology was a landmark for the development of cells that are able to perform several tasks in different areas of research, such as therapy[1], diagnosis[2], biosensing[3], bioremediation[4], and plant and animal genetic improvement[5–7]. Over the last decade, the desirable traits were mostly monogenic, with transgene expression being mainly transcriptionally controlled by tissue/development-specific or chemically/physically inducible promoters, acting as activators or repressors separately or in loops[8,9]. Currently, novel approaches in synthetic biology are being used for the design of new molecular entities that would lead to the creation and/or improvement of pathways, reprogramming organisms for new tasks[10,11]. This scenario demands the development of tools to finely tune polygenic expression.

Large serine-type phage integrases have been described as useful genomic tools for gene manipulation[12–17]. These proteins belong to a superfamily of site-specific serine recombinases, which differ in many aspects from their counterparts, the tyrosine recombinases. The latter are bidirectional proteins, which is a major drawback to their use to control gene expression due to the instability of the on/off end product[18]. Instead, the phage-encoded serine integrases (Ints) are capable of unidirectional recombination that leads to permanent DNA fragment inversion, which ultimately could be used to modulate gene expression[19]. To perform this function, Ints recognize specific attachment sites, named *attB* and *attP*[20]. When both sites are placed in opposite directions flanking a genetic part (such as a promoter, coding sequence and/or terminator), Ints perform a 180° flip of these genetic parts, which can turn gene expression on/off. As a result of this recombination event, the attachment sites are converted into different sequences, called *attL* and *attR*[21]. This recombination process can only be reversed in the presence of a cognate protein, called recombination directionality factor (RDF)[13].

Ints have been previously used to build logic gates based on Boolean algebra to regulate gene expression in prokaryotic organisms[22,23]. In these reports, the authors utilized Ints to rotate promoters or terminators, hence controlling RNA polymerase flow in a way resembling an electronic transistor, now called a transcriptor[22]. Most importantly, GFP expression for every Int used was observed as predicted by the truth table of each designed logic gate that was evaluated[22,23]. Another strategy used to regulate prokaryotic gene expression was recombinase-based state machines, a system in which several combinations of integrase inputs that inverted or excised genetic parts produced different outputs[24]. However, in eukaryotic organisms, there is a scarcity of effective tools that allow broad and precise gene regulation, which is an essential requirement for multiplex gene control. To date, only a limited number of Ints have been successfully tested in eukaryotic cells, showing a restricted cell-type-dependent functionality[25–28]. Nevertheless, most of these studies aimed to integrate or excise a DNA fragment from the genome of a particular organism, rather than using the Ints as regulators of gene expression[28–35]. In a different approach, Weinberg et al.[26] used a combination of four tyrosine and two serine recombinases in mammalian cells to develop a six-input/one-output Boolean logic lookup table. However, the relatively small number of functional Ints characterized in eukaryotic cells restricts the scale-up of the use of these proteins to build genetic circuits.

Recently, Yang et al.[18] identified 34 putative Ints, showing functionality for 11 of these Ints in prokaryotic cells. In an attempt to fill this existing gap in eukaryotic cells, we built unidirectional genetic switches to evaluate the functionality of six out of these 11 Ints that have already been tested in prokaryotic cells. To this end, we chose three different model systems: the human embryonic kidney cell lineage (HEK 293T), bovine primary fibroblasts, and *Arabidopsis thaliana* protoplasts. In addition, as a medically important proof of concept, unidirectional genetic switches using these Ints were also evaluated in peripheral blood mononuclear cells (PBMCs), neural stem cells (NSCs) differentiated from induced pluripotent stem cells, and undifferentiated human embryonic stem cells (hES, BR-1 cell line).

Here we report that all tested Ints are functional genetic switch controllers, activating the coding sequence or the promoter switches designed to be turned on in the eukaryotic cells. The frequency of cells emitting the reporter fluorescence varies among the tested integrases and cell types, and, in some cases, the switch activation is proven by molecular tests. In addition, Ints show accuracy in their site recognition and recombination process, and are not cytotoxic for the cell models assayed. These data put the evaluated Ints as suitable candidates to regulate gene expression in wide synthetic genetic networks that now can be built for several eukaryotic organisms.

## Results

**Unidirectional genetic switches design**. In our system, the HEK 293T lineage, bovine primary fibroblasts, and *A. thaliana* protoplasts were chosen as human, nonhuman mammal and plant models, respectively. Ints 2, 4, 5, 7, 9, and 13 evaluated in *Escherichia coli* by Yang et al.[18] were selected to be tested in eukaryotic switch systems. The phiC31 and Bxb1 integrases were also evaluated because they were shown to be functional in genetic switches in human cells[26] and plants[32] (only phiC31). We designed unidirectional genetic switches composed of two sets of synthesized plasmids. The first set contains either a human or plant-optimized (*A. thaliana*) coding sequence of an Int under the control of a strong species-specific constitutive promoter, named integrase expression vectors (set 1, pIE). The second set contains the reporter gene *egfp* in the reverse complement orientation flanked by the recognition sites *attB* and *attP* of the corresponding Int under distinct constitutive promoters, for either plant or animal systems. The resulting plasmids were called switch GFP vectors (set 2, pSG) (Fig. 1; Supplementary Fig. 1; Supplementary Methods). Therefore, eight plasmids were generated for each vector set of the mammalian systems, and the same number was generated for the plant system (Supplementary Table 1). One plasmid of set 1 and one plasmid of set 2 were used to transiently cotransfect mammalian cells and cotransform plant protoplasts (test condition). The negative control cells were cotransfected/cotransformed with only one of the two plasmids plus a mock plasmid to keep the DNA concentration constant. As a positive control (pGFP), the cells were cotransfected/cotransformed with a plasmid containing *egfp* in forward orientation under the same constitutive promoter as that in the plasmids from set 2 plus a mock plasmid. HEK 293T control cells were not cotransfected with the mock plasmid. In these transient assays many copies of both plasmids for each test or control conditions are inserted in the cells, according to the concentrations described in the Methods section.

The rationale for this system was that if a particular Int is functional, it would switch the *egfp* coding sequence to the forward orientation, leading to *egfp* expression and the formation of the *attL* and *attR* sites, referred to as the activated switch vector (Fig. 1; Supplementary Fig. 1). The six Ints were chosen because their *attL* sites formed after flipping did not lead to the formation of any ATG upstream of the *egfp* ORF. Although there was not ATG formation in its *attL*, Int 10 was not selected to be evaluated because its recognition sites were recognized by other integrases when evaluated in bacterial cells[18]. All assays were evaluated at 48 h for mammalian cells and 24 h for *A. thaliana* protoplasts, corresponding to the maximum EGFP-positive cell frequency.

No nuclear localization signal (NLS) was added to the Ints since previous studies showed contradictory results regarding the

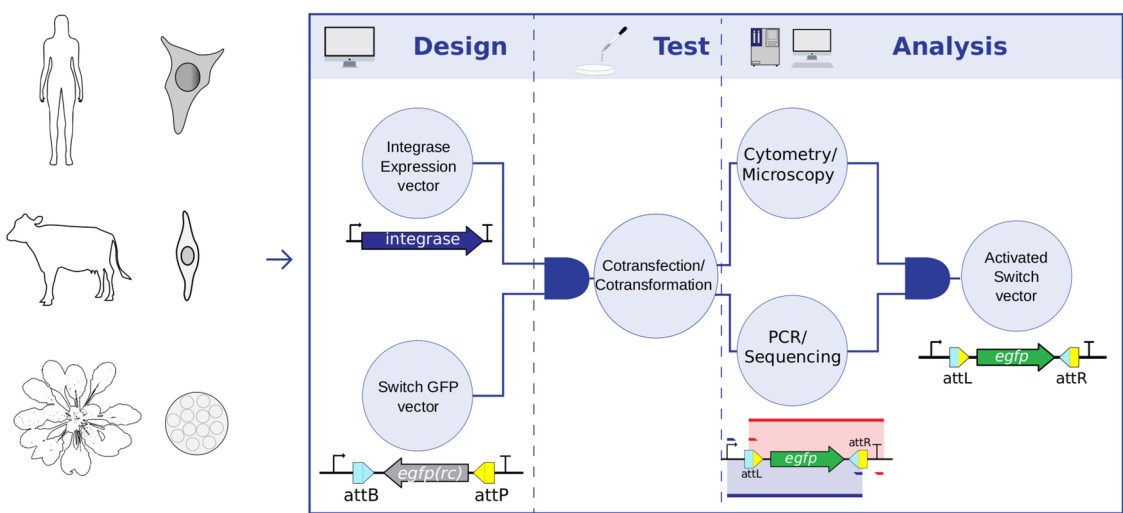

**Fig. 1 Strategy overview of the eukaryotic genetic switch system.** The human cell lineage HEK 293T, bovine fibroblasts and *A. thaliana* protoplasts were the selected model systems. The first step involved the design of two plasmid sets: the integrase expression vectors to express Ints 2, 4, 5, 7, 9, 13, phiC31, or Bxb1 and the switch GFP vectors with the *attB* and *attP* recognition sites of the respective Int flanking an *egfp* coding sequence in a reverse complement (rc) orientation. Acting as a schematic AND gate, combination of the corresponding plasmids of each of the vector sets results in the second step in the process, the test, accomplished by cotransfection or cotransformation assays of mammalian and plant cells, respectively. The third and last step led to the development of analytical methods that include the inputs of an additional schematic AND gate. Microscopy/flow cytometry analyses were used to detect EGFP fluorescence in cells resulting from the flipping action of the integrase. PCR/sequencing was used in the analysis of the *egfp* coding sequence rotated to the correct forward orientation flanked by the formed *attL* and *attR* sites. Both analytical inputs provide evidence of the activated switch vector output. The PCRs used one primer pair to amplify the complete *attL* site and the entire *egfp* coding sequence, now in the forward orientation (blue), and a second primer pair to amplify the complete *attR* site and the *egfp* sequence (red).

addition of an NLS at the N- and/or C-termini of these proteins[17,25]. In addition, in silico analysis demonstrated the existence of potential cryptic NLSs in the Ints evaluated in this study with various scores, except for Int 5, for which no NLS was predicted (Supplementary Fig. 2).

**Functional characterization of Ints as switch controllers.** In the mammalian groups, EGFP-expressing cells were detected among HEK 293T cells cotransfected with pIE + pSG vectors corresponding to Int 13, phiC31, or Bxb1 (Fig. 2a, b; Supplementary Figs. 3 and 4) and in bovine fibroblasts cotransfected with pIE + pSG vectors corresponding to Int 9, 13, or Bxb1 (Fig. 3a, b; Supplementary Figs. 3 and 5). In addition, low levels of cells emitting EGFP fluorescence could also be observed in tests activated by Ints 2, 4, and 5 in HEK 293T cells, and by Ints 2 and phiC31 in bovine fibroblasts (Figs. 2b, 3b; Supplementary Figs. 4 and 5). In general, flow cytometry analysis showed that the abundance of EGFP-positive cells ranged from 0.03 to 16.02% for the tested switches in these cell groups, indicating different levels of Int functionality (Figs. 2b and 3b). In HEK 293T cells, Kruskal–Wallis statistics corroborated that the percentages of EGFP-positive cells cotransfected with pIE + pSG corresponding to Ints 13, phiC31, and Bxb1 resulted in significant differences compared with the negative controls ($p = 1.71 \times 10^{-5}$, $1.10 \times 10^{-4}$, and $1.50 \times 10^{-4}$, respectively; Fig. 2b; Supplementary Table 2). In bovine fibroblasts, in addition to Ints 9, 13, and Bxb1 ($p = 1.10 \times 10^{-6}$, $2.04 \times 10^{-6}$, and $9.18 \times 10^{-7}$, respectively), cells cotransfected with pIE + pSG corresponding to Ints 2, 5, and phiC31 were also significantly different from the negative controls ($p = 1.87 \times 10^{-6}$, $3.08 \times 10^{-5}$, and $1.61 \times 10^{-6}$, respectively; Fig. 3b; Supplementary Table 2). Analyzing only the herein tested integrases, Int 13 led to the highest number of EGFP-positive cells in both models evaluated. The positive cell frequencies in the Int 13 tests were 33% and 16% compared with the positive controls (pGFP) for HEK 293T cells and bovine fibroblasts, respectively.

To confirm Int functionality by targeted *egfp* sequence rotation and correct *attL/attR* sites formation, PCR and sequencing analysis were performed. To this end, two pairs of primers were used. One pair annealed to the promoter sequence (forward) and to the *attR* site (reverse) (Fig. 1, blue color), and the other pair annealed to the *attL* site (forward) and to a region next to the terminator sequence (reverse) (Fig. 1, red color). Thus, PCR amplifications are expected in only activated pSG vectors. Interestingly, although EGFP fluorescence was not detected through microscopy and flow cytometry analysis of cells tested with some of the Ints, the PCR and sequencing analysis of DNA extracted from tested cells confirmed the predicted *attL/attR* sites for Ints 2, 4, 5, 7, 9, 13, phiC31, and Bxb1 and that the *egfp* had flipped to the forward orientation (Figs. 2c, 3c; Supplementary Figs. 6–9). These results confirm that all the evaluated Ints were functional, however, their efficiency may set a threshold for detection of EGFP fluorescence by flow cytometry or microscopy analysis.

An important limiting factor to the use of Ints in genetic switches is their potential cytotoxicity. Thus we measured the viability of mammalian cells after 48 h of the integrase activity using MTT assay. Neither HEK 293T cells nor fibroblasts showed marked viability impairment comparing with control groups (Figs. 2d and 3d).

For *A. thaliana* protoplasts, EGFP-expressing cells were detected after cotransformation with the pIE + pSG vectors corresponding to Int 2, 4, 7, 9, 13, phiC31, or Bxb1 (Fig. 4a, b; Supplementary Figs. 3 and 10). All Int tests led to a statistically significant frequency of EGFP-positive cells compared with the respective negative controls, although Int 5 led to a very low percentage of EGFP-expressing cells (*p*-value for Ints 2, 4, 5, 7, 9, 13, phiC31, and Bxb1 groups were $2.57 \times 10^{-11}$, $2.22 \times 10^{-7}$, $2.28 \times 10^{-6}$, $2.45 \times 10^{-9}$, $7.24 \times 10^{-7}$, $2.01 \times 10^{-8}$, $1.28 \times 10^{-6}$, and $1.07 \times 10^{-6}$, respectively; Fig. 4b; Supplementary Table 2). As observed for HEK 293T cells and bovine fibroblasts, the flow cytometry data showed a variable number of cells expressing

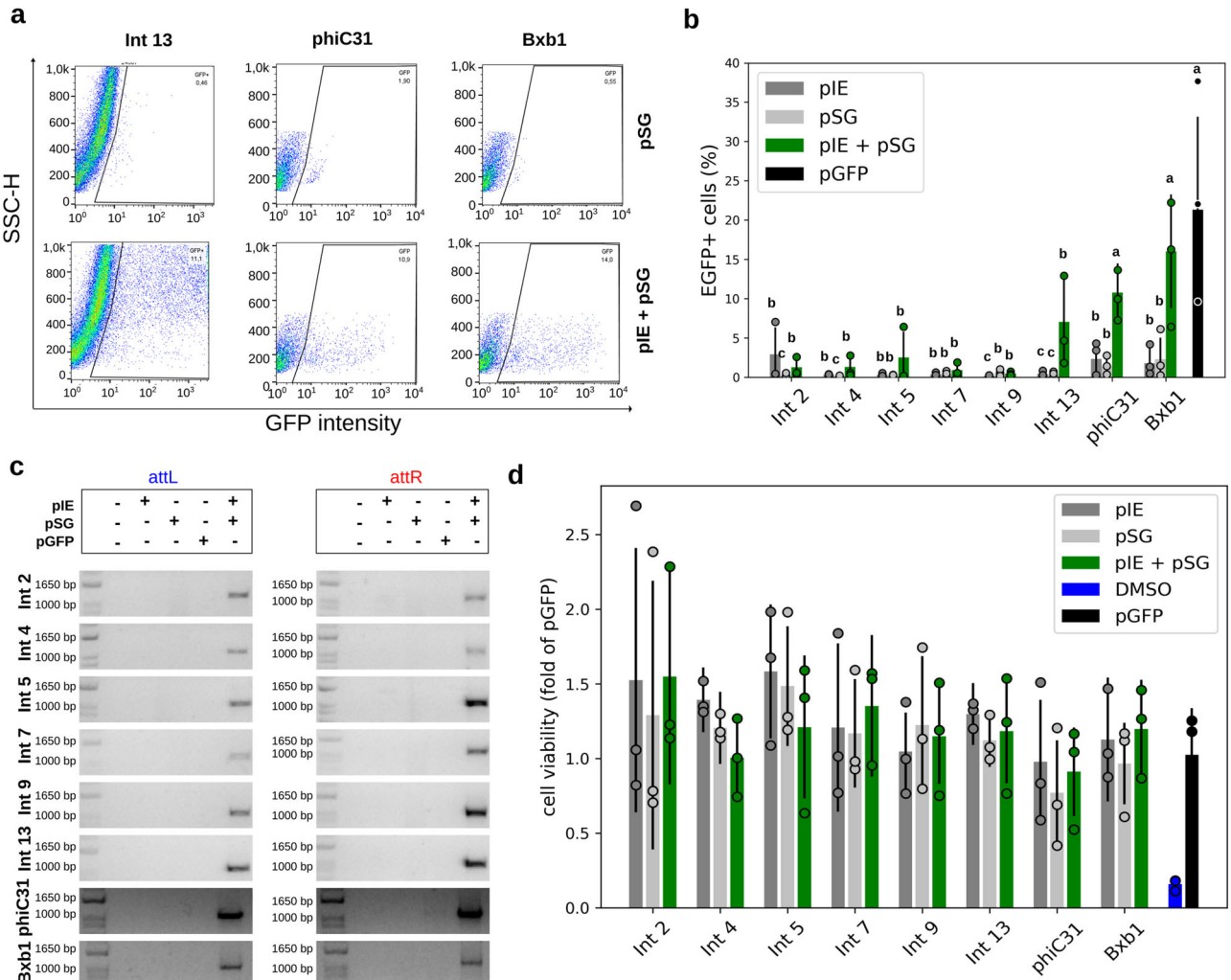

**Fig. 2 Functional characterization of the genetic switches in human cells. a** Flow cytometry distribution of HEK 293T cells at 48 h post transfection for Ints 13, phiC31, and Bxb1, the integrases that led to the highest EGFP-expressing cell frequencies. The heat map indicates the scattering of high cell concentrations (warm colors) to low cell concentrations (cool colors). The gate encompasses the EGFP-expressing cell population. **b** Bar graph plots showing the total average percentage and standard deviation of a cell population expressing EGFP in biological repeat assays ($n = 3$) and circles showing the technical duplicate or triplicate average of each assay on the y axis. The x axis contains the different conditions. For each Int data group, different letters indicate significant differences ($p < 0.05$). **c** Amplicons obtained through PCR analysis using two specific primer sets, the first set to verify *attL* formation and the second set to verify *attR* (highlighted in Fig. 1). The expected amplicon sizes in the Int test groups varied from 1021 to 1104 bp for *attL* and from 1058 to 1084 bp for *attR*. **d** Bar graph plots showing the viable cells average (circles corresponding to technical replicates averages) and standard deviation normalized with pGFP of OD measurements obtained after MTT assays ($n = 3$). DMSO corresponds to the impairment negative control. Negative control cells were transfected with only one of the two vector sets, that is, integrase expression (pIE) or switch GFP (pSG) vectors. Positive control cells (pGFP) have an *egfp* sequence in the forward orientation under the control of the EF1 alpha promoter. All the data are representative of two or three technical and three biological replicates.

EGFP. The frequency of EGFP-positive cells ranged from 0.1% for Int 5 to 24.2% for Int 13 (Supplementary Table 2). This highest value showed Int 13 to be superior to phiC31 and Bxb1 in the induction of EGFP activation. Notably, overall, plant protoplasts resulted in higher numbers of EGFP-accumulating cells than the mammalian systems tested, indicating robustness in the vegetal system (Figs. 2b, 3b, and 4b; Supplementary Table 2). In this system, the pCaMV35S-GFP plasmid, with known EGFP expression efficiency, was used as a positive control[36]. The cauliflower mosaic virus (CaMV) 35S promoter of this plasmid (alignment with GenBank V00140.1) has some single-nucleotide polymorphisms (SNPs) compared with the CaMV 35S promoter chosen for the plant switch vector syntheses (iGEM registry BBa_K1547006 and alignment with GenBank V00141.1). These SNPs, however, did not result in statistically significant differences in the frequency of EGFP-expressing cells ($p = 0.1277$; Supplementary Fig. 11).

In addition, PCR and sequencing analysis, following the same strategy previously described for the mammalian cells, showed *attL/attR* sites formation and *egfp* flipping to the forward orientation for all Ints evaluated (Fig. 4c; Supplementary Figs. 12 and 13), even for the Int 5 test condition, in which EGFP-positive cells were detected at a very low level by flow cytometry analysis.

The cell viability in protoplasts after Ints activity was measured using fluorescein diacetate (FDA) hydrolysis assay. Also, as observed in mammalian cells, in protoplasts no marked cell impairment was observed after 24 h of the integrases transformation (Fig. 4d).

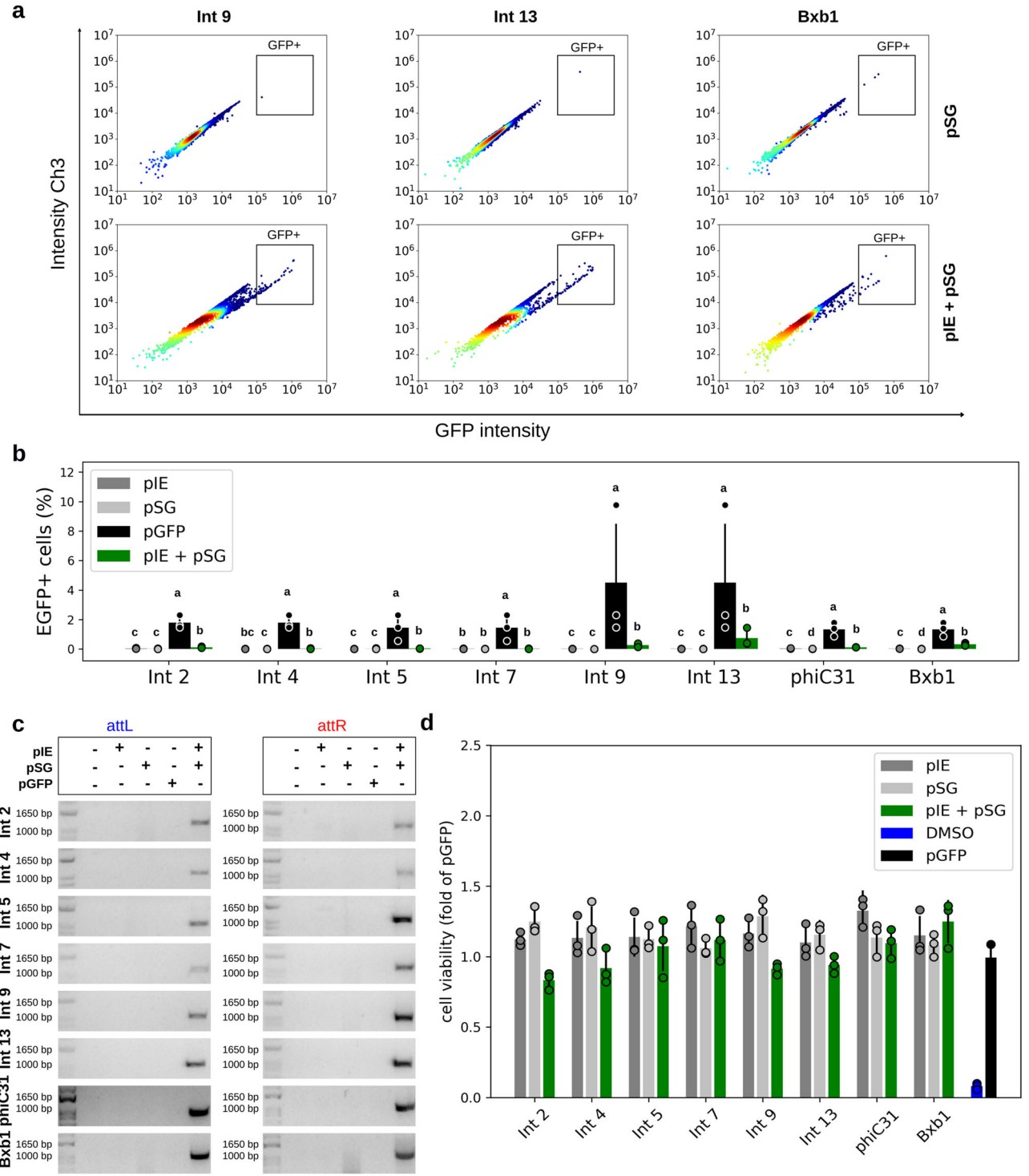

**Fig. 3 Functional characterization of the genetic switches in bovine cells. a** Flow cytometry distribution of bovine fibroblasts at 48 h post cotransfection for Ints 9, 13, and Bxb1, the integrases that led to the highest EGFP-expressing cell frequencies. The heat map indicates the scattering of high cell concentrations (warm colors) to low cell concentrations (cool colors). The gate encompasses the EGFP-expressing cell population. **b** Bar graph plots showing the total average percentage and standard deviation of a cell population expressing EGFP in biological repeat assays ($n = 3$) and circles showing the technical triplicate average of each assay on the y axis. The x axis contains the different conditions. For each Int data group, different letters indicate significant differences ($p < 0.05$). **c** Amplicons obtained through PCR analysis using two specific primer sets, the first set to verify *attL* formation and the second set to verify *attR* (highlighted in Fig. 1). The expected amplicon sizes in the Int test groups varied from 1021 to 1104 bp for *attL* and from 1058 to 1084 bp for *attR*. **d** Bar graph plots showing the viable cells average (circles corresponding to technical replicates averages) and standard deviation normalized with pGFP of OD measurements obtained after MTT assays ($n = 3$). DMSO corresponds to the impairment negative control. Negative control cells were transfected with only one of the two vector sets, that is, integrase expression (pIE) or switch GFP (pSG) vectors, plus a mock plasmid. Positive control cells (pGFP) have an *egfp* sequence in the forward orientation under the control of the EF1 alpha promoter. All the data are representative of three technical and three biological replicates.

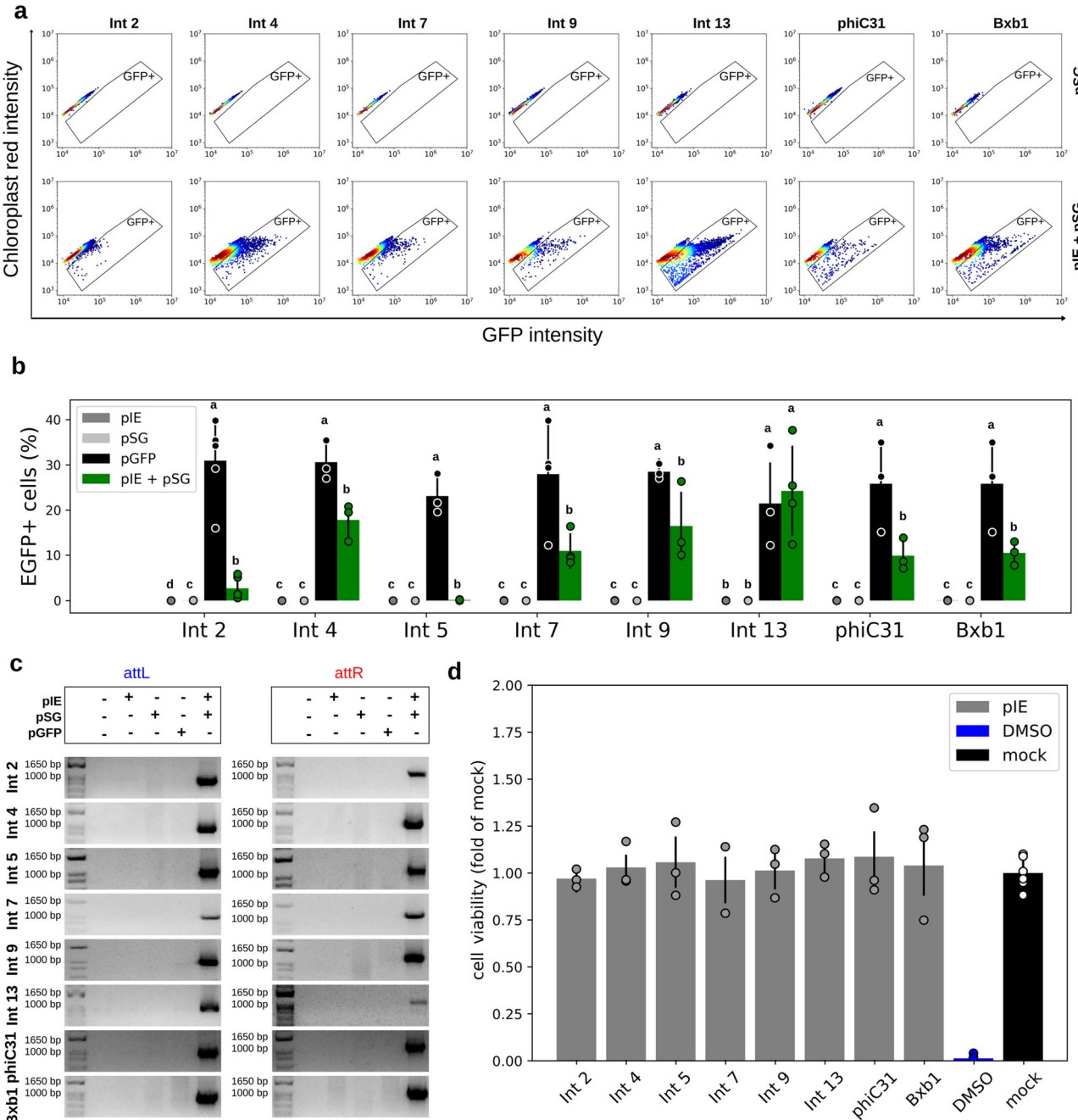

**Fig. 4 Functional characterization of the genetic switches in plant protoplasts. a** Flow cytometry distribution of protoplasts at 24 h post cotransformation for Ints 2, 4, 7, 9, 13, phiC31, and Bxb1, the integrases that led to the highest EGFP-expressing cell frequencies. The heat map indicates the scattering of high cell concentrations (warm colors) to low cell concentrations (cool colors). The gate encompasses the EGFP-expressing cell population. **b** Bar graph plots showing the total average percentage and standard deviation of a cell population expressing EGFP in biological repeat assays (n in Supplementary Table 2) and circles showing the technical triplicate average of each assay on the y axis. The x axis contains the different conditions. For each Int data group, different letters indicate significant differences ($p < 0.05$). **c** Amplicons obtained through PCR analysis using two specific primer sets, the first set to verify *attL* formation and the second set to verify *attR* (highlighted in Fig. 1). The expected amplicon sizes in the Int test groups varied from 948 to 983 bp for *attL* and from 1136 to 1132 bp for *attR*. Negative control cells were transfected with only one of the two vector sets, that is, integrase expression (pIE) or switch GFP (pSG) vectors, plus a mock plasmid. Positive control cells (pGFP) have an *egfp* sequence in the forward orientation under the control of the CaMV 35S promoter. All the data are representative of three technical and three or more biological replicates. **d** Bar graph plots showing the viable cells average (circles corresponding to technical replicates averages) and standard deviation normalized with mock plasmid positive cells obtained after FDA assays. DMSO corresponds to the impairment negative control. The cells were transfected with pIE vectors in technical triplicates and three biological replicates (except for Int 7, for which there was three technical and two biological replicates).

Another important concern related to the use of Ints is the accuracy with which the Ints mediate the site-specific recombination. Therefore, the individual sequence clone reads, obtained from PCR amplicons, were aligned with the expected flipping plasmid sequence, trimmed, and analyzed for all three cell model systems evaluated. In addition to confirming Int functionality, DNA sequencing demonstrated minimal errors in the recombined *attL/attR* sites, indicating that these proteins are not error prone. For only Int 4 in the bovine fibroblast system and Int 7 in the protoplast system, one covered SNP was observed at the *attR* sites. However, mutations were also observed in the *egfp* sequence for most clones sequenced, suggesting that the observed mutations may be due to PCR errors instead of imprecise Int recombination (Supplementary Table 3; Supplementary Data 1, 2, 3).

Sequencing analyses also evidenced a one-base difference for the Int 9 *attL/attR* attachment sites compared with the original described by Yang et al.[18] (Supplementary Fig. 14). These data indicate that the adenine after the previously marked *attB* core and a thymine after the *attP* should be part of the Int 9 crossover site. However, this arrangement would lead to a one-base difference between the *attB/attP* cores, making it impossible for dinucleotide overhang recombination to occur after integrase subunit rotation[37]. Thus, we found a possible arrangement with the next CT dinucleotide as the core for the Int 9 attachment sites.

**Promoter as switchable genetic part and Ints orthogonality.** We wanted to evaluate whether a different genetic component could be used as a switchable part and if these proteins were orthogonal. To this end, *A. thaliana* protoplast was chosen as a model organism, since we observed a high number of cells expressing EGFP in this system, indicating robust Int activity. Three Ints with variable levels of efficiency were selected: Ints 2, 4, and 5. Although Int 13 led to the highest *egfp* activation in protoplasts, it was not selected because there is an ATG start codon in the *attR* site formed after recombination, what would remove the *egfp* gene sequence frame, and because Yang et al.[18] reported a constitutive promoter activity for Int 13 *attP* site.

In this proposal, the switch GFP construct (set 2 plasmid) was redesigned, placing the constitutive promoter, CaMV 35S, in the reverse complement orientation flanked by the *attB* and *attP* sites of Ints 2, 4, and 5 in tandem, followed by the *egfp* sequence in the forward orientation; this construct was named the switch promoter vector (pSP) (Fig. 5a). Flow cytometry data showed 32.2%, 38.6%, and 12.5% EGFP-positive cells in the protoplast population cotransformed with the pIE Int vectors 2, 4, and 5 + pSP, respectively (Fig. 5b, c, Supplementary Figs. 3 and 15). Residual EGFP accumulation on the pSP negative control cells was observed; however, all three Int tests exhibited statistically significant differences compared with the negative control groups ($p = 1.51 \times 10^{-11}$; Supplementary Table 4). Notably, a higher EGFP-positive cell frequency was observed for the Int 2 test than for the flipping strategy with the switch GFP vector.

PCR and sequencing analysis corroborated these results, showing *attL/attR* sites formation and CaMV 35S flipping, driving the RNA polymerase to transcribe the *egfp* gene (Fig. 5d; Supplementary Fig. 16). Because we used primer sets flanking the recognition sites of Ints 2, 4, and 5 (Fig. 5a, activated switch promoter vector), we could observe that each Int only recognized its own *attB/attP* sites (Supplementary Fig. 17), with just one covered deletion observed at the formed *attL* site of Int 5 (Supplementary Table 5, Supplementary Data 4). These results indicate that Ints 2, 4, and 5 are accurate and orthogonal. In conclusion, the promoter sequence can also be used to build genetic switches.

**Ints activity in primary human T lymphocytes and stem cells.** As a proof of concept, unidirectional genetic switches using integrase expression (pIE) and switch GFP (pSG) vectors were tested in human peripheral blood mononuclear cells (PBMCs), the main source of T lymphocytes. These cells were chosen due to their significance in several different areas of health research, such as vaccine development, hematological malignancies, high-throughput screening, and immunology. Recently, T lymphocytes have gained importance in cancer immunotherapies[38,39] through their clinical use with or without genetic manipulation. Considering the results obtained for HEK 293T cells, we selected Int 13, an integrase with highly detectable functionality, and Int 4, which showed a low frequency of EGFP-expressing cells, to be tested with PBMCs; phiC31 and Bxb1 were also evaluated. Flow cytometry analysis showed that the Int 13 and Bxb1 tests resulted in the same EGFP-expressing cell frequency (7%) in PBMCs extracted from three healthy independent donors (Fig. 6a, b; Supplementary Fig. 18). These results indicate that Int 13 and Bxb1 were able to promote the inversion of the *egfp* coding sequence to the forward orientation, leading to that high EGFP-expressing cells frequency, equivalent to 87.5% of the result observed in the positive control population (pGFP).

PCR and sequencing analysis performed using the same strategy previously described for the other switch GFP assays corroborated these data and expanded the data, showing that Ints 4 and phiC31 were also capable of flipping the *egfp* coding sequence (Fig. 6c; Supplementary Figs. 19 and 20). The DNA sequence reads obtained were also aligned with the activated plasmid expected sequence, and the covered mutations were counted. Only a few SNPs were observed at the formed *attL/attR* sites (Supplementary Table 6, Supplementary Data 5).

In addition, this study aimed to investigate Int functions in another human cell type used as an efficient in vitro model in studies on several diseases and embryogenesis. Switch systems using integrase expression vectors (pIE Ints 2, 9, 13, phiC31, and Bxb1) and the respective switch GFP (pSG) vectors were thus evaluated in neural stem cells (NSCs), and undifferentiated human embryonic stem (hES) cells. The plasticity of these cells has made them the focus of basic developmental research and also of the challenging regenerative medicine field. As an example, NSCs exhibit promise in the treatment of neurodegenerative diseases[40]. The hES-BR-1 cell line was the first hES cell line established from the Brazilian population and is a relevant model for stem cell assays[41]. The flow cytometry measurements were within a narrow range, with 2.0–3.9% EGFP-positive cells observed in the Int 2, 9, and 13 tests (Fig. 7), indicating that these Ints are functional in these relevant disease modeling systems. PhiC31 and Bxb1 tests exhibited the highest values, showing activity in between 9.6 and 24.8% of EGFP-positive cells (Fig. 7).

**Discussion**
Although tools based on activators and repressors, including recently modified CRISPR-Cas9 systems[42], are used to regulate eukaryotic gene expression, there remains a need for new technologies that are scalable and precise for multiplex gene regulation control. In our study, through eukaryotic cell-based assays, we demonstrated the widespread use of six Ints 2, 4, 5, 7, 9, and 13 as genetic switchers in mammalian and plant cells. All the Ints tested were able to (i) recognize their respective predicted *attB/attP* sites, (ii) generate the predicted *attL/attR* sites, and (iii) invert the *egfp* or promoter sequences. Furthermore, our results demonstrated that these six Ints have different degrees of functionality depending on the eukaryotic cell type or the genetic part to be flipped, potentially leading to a tunable system. Of the six Ints evaluated, the flow cytometry analysis showed that Int 13 led

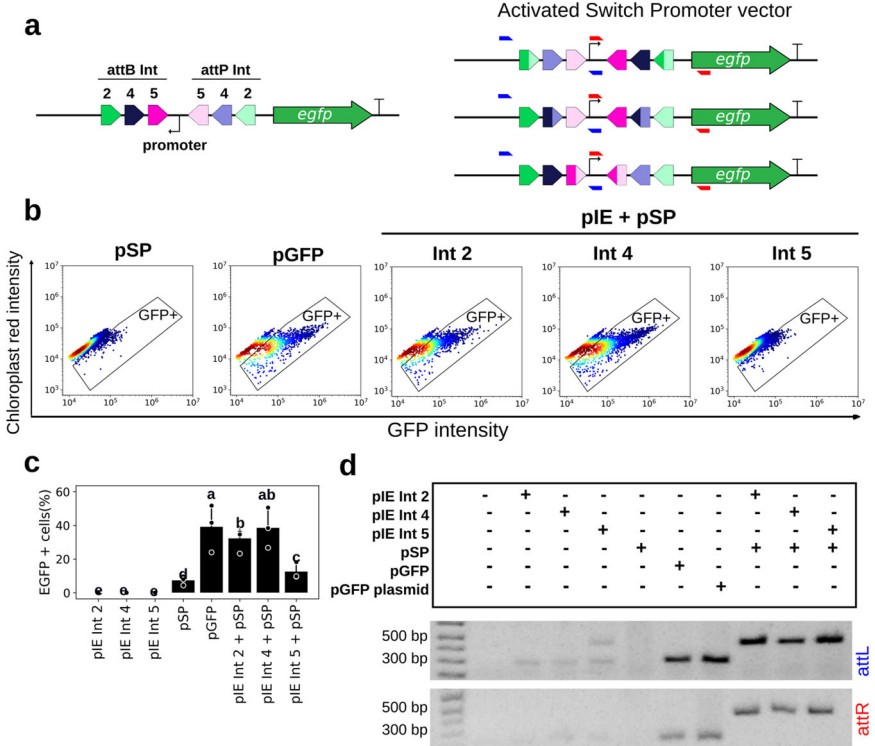

**Fig. 5 Promoter as a switchable genetic part in plant protoplasts. a** Schematic representation of the CaMV 35S promoter as a switchable genetic part. In this case, the switch vector was redesigned to contain the promoter in the reverse complement sequence orientation flanked by the *attB* and *attP* sites of three different Ints (2, 4, and 5) in tandem; this construct was named the switch promoter vector (pSP). The integrase expression vectors (pIE) for each Int were cotransformed separately with the pSP, and each resulting activated promoter vector is shown. **b** Flow cytometry distribution of protoplasts at 24 h post cotransformation. The heat map indicates the scattering of high cell concentrations (warm colors) to low cell concentrations (cool colors). The gate encompasses the EGFP-expressing cell population. **c** Bar graph plots showing the total average percentage and standard deviation of a cell population expressing EGFP in biological repeat assays ($n = 3$) and circles showing the technical triplicate average of each assay on the *y* axis. The *x* axis contains the different conditions. Different letters indicate significant differences ($p < 0.05$). **d** Amplicons obtained through PCR analysis using two specific primer sets. The first primer set (blue) was complementary to the pSP vector backbone sequence (forward) and promoter sequence (reverse). Expected amplicon size: 429 bp. The second primer set (red) was complementary to the promoter sequence (forward) and egfp coding sequence (reverse). Expected amplicon size: 438 bp (primers are colored and marked in letter a). Negative control cells were cotransformed with one of the integrase expression vectors (pIE) plus a mock plasmid or with the switch promoter vector (pSP) plus a mock plasmid. Positive control cells (pGFP) have a plasmid containing the *egfp* sequence under the control of the CaMV 35S promoter in the forward orientation plus a mock plasmid. Expected amplicon size for pGFP: *attL* gel: 285 bp; *attR* gel: 227 bp. These amplicons were smaller than those obtained under the test conditions due to the absence of the Int attachment sites. All the data are representative of three technical and three biological replicates.

to the highest proportion of EGFP-expressing cells in all three eukaryotic model systems. Int 13 (NCBI No. WP_012095429.1) is a protein with 55.5 kDa in a range of 54.0–67.1 kDa among the eight integrases evaluated, and it was identified from a prophage inside *Bacillus cytotoxicus* NVH 391-98[18,43]. Int 9 for bovine fibroblasts and Ints 4 and 9 for plant protoplasts were set at the second position. PhiC31 and Bxb1, characterized elsewhere, yielded the highest EGFP percentage values only for HEK 293T cells, slightly overcoming Int 13 activity. On the other hand, for bovine fibroblasts, Bxb1 exhibited activity close to that of Int 9, and phiC31 showed low activity. In protoplasts, these integrases promoted an intermediate effect, similar to that of Int 7. Ints 9, 4, and 13 yielded higher percentages of EGFP-positive cells than phiC31 and Bxb1 in this plant model. Moreover, interestingly, in the study conducted by Yang et al.[18] in bacteria, Ints 2, 4, 5, 7, and 13 yielded 100% GFP-positive cell populations and Int 9 yielded ~80%. In eukaryotic model systems, different inherent factors of these highly complex organisms can be hypothesized to interfere with integrase functionality, leading to different overall results. However, this variability of Ints potency in eukaryotic systems can be used to design genetic circuits with distinct functionalities and modulation proprieties.

Despite none of these Ints were previously evaluated as regulators of gene expression in eukaryotic cells, two studies evaluated different Ints performing other functions in yeast (*Saccharomyces cerevisiae*) and in two mammalian cell types[25,29]. These authors showed that 10 out of 14 integrases were active in *S. cerevisiae* through recombinase exchange reactions and that 7 out of 15 Ints promoted site-specific deletions in both lineages of mammalian cells. They also showed that seven Ints were cytotoxic in yeast and that all 15 were cytotoxic in mammalian cells to some degree, and some of the Ints were error prone in the mammalian systems. Here, the *attL*/*attR* sites were amplified and sequenced, and no mutations in these sites were found for Ints 2 and 9 in any of the models tested. Regarding the other Ints, only a few mutations were observed in some models. In addition, our cell viability assays indicated no limiting toxicities related to the integrases activity, suggesting that the systems reported herein can be used in stable expression-based experiments.

Taking into account the CaMV 35S promoter-based switch construction for protoplasts, despite a residual leak observed in the switch promoter vector negative control, Ints 4 and 5 led to a result that was compatible with the switch GFP system. Int 2, however, exhibited a much higher number of EGFP-positive cells

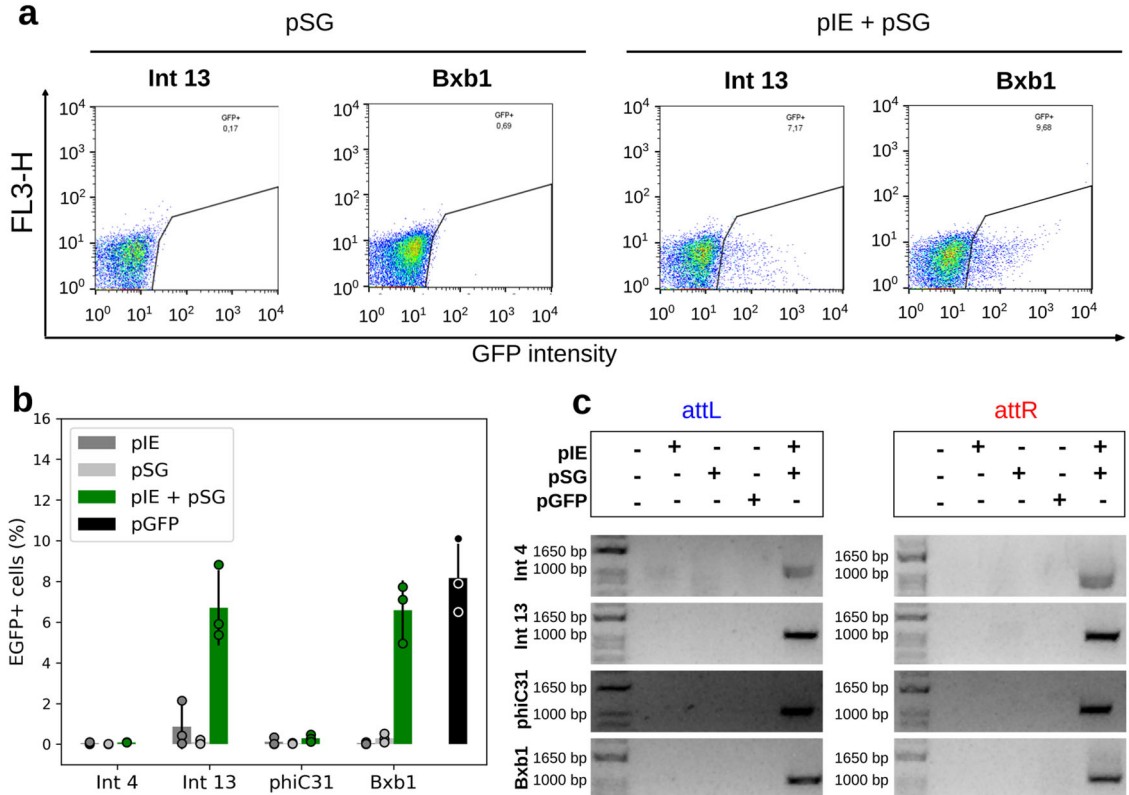

**Fig. 6 Int activity in primary human T lymphocytes from PBMCs isolated from three independent donors. a** Flow cytometry distribution of PBMCs at 48 h post electroporation for GFP switch tests with Int 13 and Bxb1, the integrases that led to the highest EGFP-expressing cell frequencies. The heat map indicates the scattering of high cell concentrations (warm colors) to low cell concentrations (cool colors). The gate encompasses the EGFP-expressing cell population. **b** Bar graph plots showing the total average percentage and standard deviation of a cell population expressing EGFP in biological repeat assays ($n = 3$) and circles corresponding to single data points from each donor material. The x axis contains the different conditions. In the PBMCs assays, Ints 2, 13, phiC31, and Bxb1 were evaluated. **c** Amplicons obtained through PCR analysis using two specific primer sets, the first set to verify attL formation and the second set to verify attR (highlighted in Fig. 1). The expected amplicon sizes in the Int test groups varied from 1021 to 1104 bp for attL and from 1058 to 1084 bp for attR. Negative control cells were electroporated with only one of the two vector sets, that is, integrase expression (pIE) or switch GFP (pSG) vectors. Positive control cells (pGFP) have an egfp sequence in the forward orientation under the control of the EF1 alpha promoter. All the data were representative of three donors, corresponding to biological triplicates with single measurements.

when the promoter was flipped than when the egfp sequence was. Moreover, this system suggested that these three Ints are orthogonal, as previously demonstrated in bacteria[18].

Therefore, these proteins can be multiplexed in different combinations of genetic switches or logic gates based on Boolean algebra to build genetic circuits. Such applications would facilitate differential control of metabolic routes or synthetic transgenic systems in livestock animals or crop plants, allowing precise and efficient regulation of gene expression regarding, for example, abiotic/biotic conditions or growth timing responses to activate or deactivate resistance, defense, or nutritional improvement genes.

An important challenge to be considered in studies that aim to develop biotechnological tools with wide applicability is proof of concept. Here, in addition to the evaluation of Ints in model systems, unidirectional genetic switches were evaluated in cells of great clinical relevance, such as primary T lymphocytes, and stem cells[39,44,45]. The therapies based on T lymphocytes and stem cells represent a field of research with a demand for transcription systems with fine-tuning capabilities[46], such as the Int-based systems described in this study. As for the model systems HEK 293T cells, bovine fibroblasts, and plant protoplasts, our results in PBMCs, NSCs, and hES cells showed that the evaluated Ints were able to recognize the attB/attP sites, precisely forming the attL/attR sites and performing the 180° rotation (flip) of the egfp coding sequence. Accordingly, not all the Ints tested led to high

frequencies of EGFP detection by flow cytometry and microscopy analysis, indicating differential Int activity in these cells. These results are very relevant because, first, we were able to show that these phage proteins are active in eukaryotic cells used in a variety of biologically important studies, reinforcing the robustness of the Int platform[47]. Second, we can foresee the construction of genetic circuits to improve the already successful cancer immunotherapy strategies and for a wide range of potential applications of Ints in disease modeling in vitro and in therapeutic-based approaches in human stem cells. One important contribution in this context can be using Ints to refine specific temporal/space gene activation/deactivation, minimizing potential undesired side effects.

The use of Ints can also be expanded to the study of essential genes in eukaryotic organisms. Recently, Cre/Lox tyrosine integrases were utilized in a SCRaMbLE analysis to identify nonessential genes from chromosome III of S. cerevisiae[48]. However, due to the use of only one integrase, all genes called nonessential were extracted from the genome of this organism at the same time, resulting in immediate loss of cell viability due to unknown genome redundancy[48]. Our work led to a considerable increase in the number of functionally characterized Ints that were available for use in eukaryotic cells. Now, these Ints can be multiplexed to flank several endogenous genes in random combinations, making it possible to knockout separate groups of genes. Ultimately, this multiplex strategy can allow the investigation of functional gene

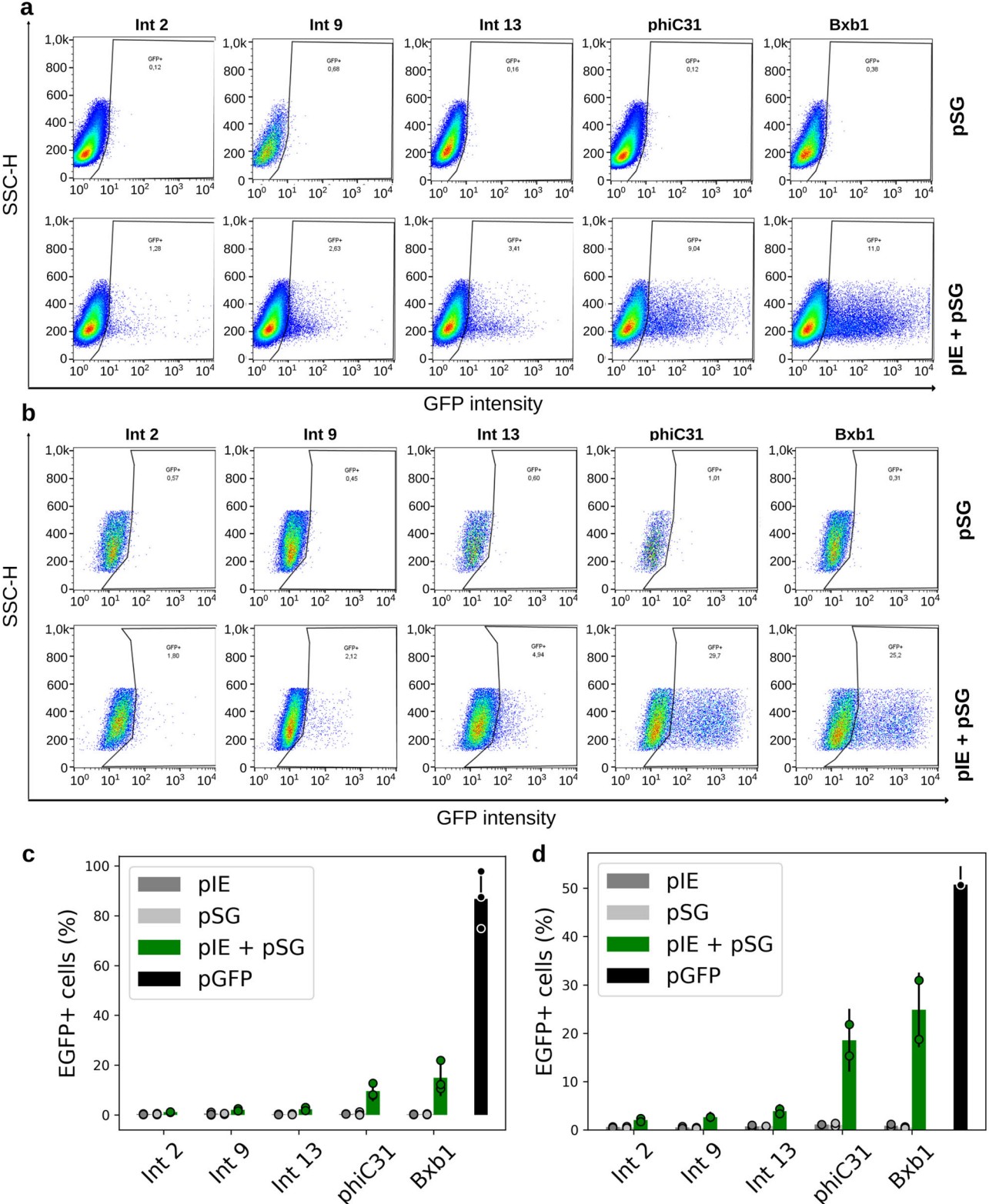

**Fig. 7 Int activity in stem cells.** Flow cytometry distribution of NSCs (**a**) and hES cells (**b**) at 48 h post electroporation for GFP switch tests with Ints 2, 9, 13, phiC31, and Bxb1. The heat map indicates the scattering of high cell concentrations (warm colors) to low cell concentrations (cool colors). The gate encompasses the EGFP-expressing cell population. Bar graph plots showing the total average percentage and standard deviation of NSCs (**c**) and hES cell (**d**) populations expressing EGFP in biological repeat assays ($n = 3$ for NSCs; $n = 2$ for hES cells) and circles showing the technical triplicate average of each assay on the $y$ axis. The $x$ axis contains the different conditions. In the stem cell assays, Ints 2, 9, 13, phiC31, and Bxb1 were evaluated. Negative control cells were transfected with only one of the two vector sets, that is, integrase expression (pIE) or switch GFP (pSG) vectors. Positive control cells (pGFP) have an *egfp* sequence in the forward orientation. All the data are representative of three technical and three (NSCs) or two (hES cells) biological replicates.

redundancy on the genome-scale. Furthermore, as different Ints have different degrees of functionality, these proteins can be used in studies of vital multifamily genes, allowing modulation of gene expression, or even to evaluate extrachromosomal toxic protein-coding genes. Due to their accuracy, Ints can also be used to investigate the roles of specific domains of selected genes or gene families, flanking predicted functional domains with their attB/attP sites and triggering the flipping of these sequences to the nonfunctional reverse orientation in a specific tissue or developmental stage. Finally, these proteins can also be used as DNA barcodes to identify cell lineages in developmental and evolutionary studies[49].

Importantly, in this study, six Ints were evaluated, so the number of functional Ints in eukaryotic cells can be substantially increased by taking into account the pool of more than 4000 integrases observed by Yang et al.[18] in sequence databases with predicted recognition sites for 34 of these proteins.

Furthermore, studies regarding nuclear localization and protein accumulation could be performed, leading to improvement in Int functionality with the addition of NLSs or removal/modification of putative degradation signals. Last, the results presented in this work indicate that Ints 2, 4, 5, 7, 9, and 13 can further be used for a myriad of biotechnological applications.

## Methods

**Integrases and plasmids.** The serine integrases 2, 4, 5, 7, 9, 13[18], phiC31[50], and Bxb1[22] were codon optimized by an online codon optimization tool (IDT software) for expression compatibility in the respective eukaryotic system: for mammalian cells, the proteins were codon optimized for expression in *Homo sapiens*, and for protoplasts, the proteins were codon optimized for expression in *A. thaliana*. Two plasmid sets were synthetically constructed (Epoch Life Science Inc.) to compose the genetic switch systems. The first plasmid set was constructed to express the Int gene (integrase expression vectors (pIE)). The second plasmid set, the switch vectors, carried the reporter *egfp* gene (switch GFP vectors (pSG)) or its promoter (switch promoter vector (pSP)) in reverse complement orientation, flanked by the attB site and the reverse complement sequence of the attP site of the Int expressed by the first plasmid. Addgene accession numbers of all vectors used in this study are described in Supplementary Table 1. Vector's part sequences are also provided in Supplementary Methods.

**Mammalian system plasmids.** For the integrase expression vectors, Ints 2, 4, 5, 7, 9, 13, phiC31, and Bxb1 were placed under the control of the ubiquitin promoter and β-globin poly(A) signal terminator. The coding sequences of these integrases were cloned into the pUB-GFP plasmid (Addgene, 11155), replacing the GFP-coding sequence, which resulted in a set of integrase-expressing vectors called pUB-HspINTX (X = 2, 4, 5, 7, 9, 13, phiC31, or Bxb1). For the switch GFP vectors, the *egfp* coding sequence (Addgene, 11154) was cloned in reverse complement orientation, flanked by the attB site and the reverse complement sequence of the attP site of the individual Ints 2, 4, 5, 7, 9, 13, phiC31, and Bxb1. These cassettes were cloned into the pEF-GFP plasmid (Addgene, 11154), replacing the original *egfp* coding sequence under the regulation of the EF1 alpha promoter and β-globin poly(A) signal terminator. The resulting switch GFP vectors were named pEF-GFP (rc)X (X = 2, 4, 5, 7, 9, 13, phiC31, or Bxb1) (Supplementary Fig. 1a). The GFP expression-positive control pT3-Neo-EF1α-GFP (Addgene, 69134) was used for HEK 293T and PBMCs assays; the pEF-GFP plasmid (Addgene, 11154) was used for bovine fibroblast assays; and the pT2-GFP (previously kindly provided[51]) was used for NSCs and hES cell assays.

**Plant system plasmids.** In this system, the Ints 2, 4, 5, 7, 9, 13, phiC31, and Bxb1 CDS, under the control of the actin2 gene promoter[52] and NOS terminator (pBI426 plasmid[53]), were cloned into the pUC57, pSB3K3 or pBluescript II SK(−) vectors by Epoch Life Science Inc. These plasmids resulted in a set of integrase expression vectors, individually called pAct-AtINTX (X = 2, 4, 5, 7, 9, 13, phiC31, or Bxb1). The *egfp*[36] coding sequence in a reverse complement orientation, flanked by attB and the reverse complement sequence of the attP attachment site of the Ints 2, 4, 5, 7, 9, 13, phiC31, and Bxb1, was placed under the control of the CaMV 35S promoter (iGEM registry BBa_K1547006) and the NOS terminator, constituting a set of switch GFP vectors for plants. These cassettes were inserted into the plasmids pUC18 or pBluescript SK(−) (Epoch Life Science Inc.), resulting in p35S-GFP(rc)X (X = 2, 4, 5, 7, 9, 13, phiC31, or Bxb1) (Supplementary Fig. 1b). For the plant system, a second switch vector was constructed. This vector consisted of the reverse complement of the CaMV 35S promoter flanked by the attB and the reverse complement sequence of the attP sites of the Ints 2, 4 and 5, sequentially positioned together (Fig. 5a). This promoter cassette was synthesized (Epoch Life Science Inc.)

and cloned, replacing the CaMV 35S sequence in the positive control vector pCaMV35S-GFP[36]. The final vector was called p35S(rc)2_4_5-GFP and was used as the switch promoter vector (pSP).

**HEK 293T maintenance and PBMCs isolation.** The human embryonic kidney cell lineage HEK 293T (a gift from Dr. Elio Vanin of St. Jude Children's Research Hospital) was cultivated in 75-cm² tissue culture flasks with 15 ml of Dulbecco's modified Eagle's medium (DMEM; Gibco), 10% fetal bovine serum (FBS; HyClone), and penicillin-streptomycin (10 U/ml; Gibco). The cells were detached and seeded every 2–3 days at below 80% confluence. For PBMCs, white blood cells from healthy blood donors were collected using a leukocyte reduction filter (RS; Haemonetics) and washed with phosphate-buffered saline (PBS). To isolate PBMCs, a density gradient centrifugation was performed using Ficoll-Hypaque®-1077 (GE HealthCare) (deceleration off; centrifugation for 20 min at 800 × $g$) followed by three washes with PBS. The use of PBMCs from healthy donors was approved by an Institutional Review Board (the Brazilian National Cancer Institute (INCA) Ethics Committee), and donors signed review board-approved informed consent forms.

**Human stem cell culture.** Neural stem cells (NSCs) differentiated from induced pluripotent stem cells (iPSCs)[54] and pluripotent human embryonic stem cells (hES, BR-1 cell line)[41] were used. All human stem cell experiments were approved by the ethics committee of Copa D'Or Hospital (CAAE number 60944916.5.0000.5249, approval number 1.791.182). The cells were cultured/maintained in neural advanced DMEM/F12 and neurobasal medium (50% v/v) plus neural induction supplement (NIS) medium (all from Thermo Fisher Scientific), called NEM (neural expansion medium), over Geltrex (Thermo Fisher Scientific) at 37 °C in 5% CO₂ as previously described[54,55]. hES cells were cultured in mTeSR medium (Stem Cell Technologies) over Geltrex.

**HEK 293T transfection.** A total of 4 × 10⁶ HEK 293T cells were plated in 75-cm² flasks with 15 ml of DMEM (Gibco), 10% FBS (HyClone), and penicillin-streptomycin (10 U/ml; Gibco). After 24 h, the medium was removed, and 10 ml of fresh DMEM/FBS was added. In the next step, 5 µg of each integrase expression vector pUB-IntX (X = 2, 4, 5, 7, 9, 13, phiC31, or Bxb1) and each switch GFP vector pEF-GFP(rc)X carrying the respective integrase site were added to 500 µl of 2X CaCl₂ at 250 nM. Next, 500 µl of HBS (pH 7.1) was slowly added while the solution was vortexed at 10,000 rpm. Bubbles were produced in the solution with a Pasteur pipette and mixed. The solution was incubated for 10 min at room temperature and then dripped with the Pasteur pipette throughout the flask. The medium was changed after 16 h. The results were analyzed 48 h after the transient transfection.

**PBMC, NSC, and hES electroporation.** Cells (1 × 10⁷ PBMCs, 1 × 10⁶ NSCs or hES cells) were transferred to a sterile 0.2-cm cuvette (Mirus Biotech®) and electroporated as previously described[51]. Briefly, PBMCs and hES cells were resuspended in 100 µl of 1SM buffer, and NSCs were resuspended in 100 µl of 1 S buffer. PBMCs were electroporated with 5 µg of each integrase expression (pIE) and switch GFP (pSG) plasmids using the U-14 program of the Lonza® Nucleofector® II electroporation system. NSCs and hES cells were electroporated with 12 µg of pIE plasmids and 8 µg of pSG plasmids using A-33 and A-23, respectively, from the Lonza® Nucleofector® II electroporation system. The mock control was electroporated with 100 µl of 1 SM (PBMCs and hES cells) or 1S (NSCs) buffer without plasmid (used to set flow cytometry gates). After transfection, PBMCs were gently resuspended in 1 ml of prewarmed RPMI medium supplemented with 2 mM L-Glu and 20% fetal calf serum (FCS; Gibco). NEM was used for NSCs, and mTeSR was used for hES cells. All cells were cultured for 48 h after electroporation for transient transfection, and then, the analyses were performed.

**Bovine fibroblast isolation.** Fibroblasts were isolated according to the protocol described by Freshney[56] with some modifications. The cells were removed from 14-month-old Nelore (*Bos indicus*) bull oxtail by biopsies and washed three times in 0.05% trypsin (Gibco). The cells were then transferred to 25-cm² cell culture flasks and incubated in DMEM (Gibco) supplemented with 10% FBS (Gibco) and penicillin-streptomycin at 37 °C in a 5% CO₂ atmosphere. After three passages, or when the fibroblast cultures showed homogeneity, the cells were ready for transfection. Cell cultures with 60–70% confluence were picked. The use of the bovine cells was approved by the Ethics Committee on the Use of Animals (CEUA) of Embrapa Genetic Resources and Biotechnology in March 2013 under the reference number 001/2013.

**Bovine fibroblast cell transfection.** After growth, the cells were enzymatically dissociated with a trypsin solution (0.5% trypsin, 0.2% EDTA), and after 10 min, the reaction was inactivated using DMEM (Gibco). The cells were counted in a Neubauer chamber, transferred (10⁵ cells/well) to 24-well culture dishes and grown for 24 h, when they reached >70% confluence. Primary bovine fibroblasts were cotransfected with 350 ng of each of the two plasmid sets utilizing Lipofectamine LTX & Plus Reagent (Invitrogen) and cultured in Opti-MEM (Invitrogen)

according to the manufacturer's instructions. The results were analyzed 48 h after transient transfection.

**Protoplast isolation**. The protoplasts were obtained following the protocol described by Yoo et al.[57] with some modifications. *A. thaliana* ecotype Columbia was grown under a 12-h light/12-h dark cycle at 22 °C. Four to five weeks after seeding, ~20 young leaves were collected, transferred to a plate with W5 solution (154 mM NaCl, 125 mM CaCl₂, 5 mM KCl, 2 mM MES, pH 5.7), and scalped by using a blade. Then, the leaves were placed on a digestion plate containing 5 ml of enzyme solution [0.5 M mannitol, 20 mM KCl, 20 mM MES (pH 5.7), 0.2% pectolyase (Sigma-Aldrich), 0.5% driselase (Kyowa Hakko Bio Co., Ltd.), 1.5% cellulase (Sigma-Aldrich), 10 mM CaCl₂, 1 mg/ml BSA]. A 15–20 pol Hg vacuum was applied three times for ~5 s, and the plate was incubated at room temperature in a platform shaker at 40 rpm for 3 h. The digested sample was filtered through a 44-μm mesh, and the W5 solution was added to wash the obtained protoplasts, followed by centrifugation at $100 \times g$ for 2 min. After two additional washing steps followed by centrifugation with 10 ml of W5, the protoplasts were resuspended in 1 ml of MMg solution (0.4 M mannitol, 15 mM MgCl₂, 4 mM MES, pH 5.7), and the concentration was adjusted to $4–5 \times 10^5$ protoplasts/ml.

**Protoplast transformation**. Cotransformation was performed in a 15-ml Corex tube using 100 μl of $4–5 \times 10^5$ protoplasts/ml, 10 μg of each desired plasmid DNA of the two sets of vectors and 110 μl of 40% PEG solution [PEG 4000 (Sigma), 0.2 M mannitol, 100 mM CaCl₂] for each reaction (scaling up to 6 reactions per tube). After 15 min, the reaction was stopped with two volumes of W5 solution, centrifuged at $100 \times g$ for 2 min, resuspended in 500 μl of W1 solution (0.5 M mannitol, 20 mM KCl, 4 mM MES, pH 5.7) (each reaction) and plated on a 12-well plate. The plates were incubated at room temperature in a platform shaker at 40 rpm, and the results were analyzed after 24 h transient transformation.

**Flow cytometry**. HEK 293T cells, NSCs, hES cells, bovine fibroblasts, and *A. thaliana* protoplasts were analyzed by flow cytometry in technical triplicates. PBMCs were analyzed in one sample for each of the three donors. HEK 293T cells, PBMCs, NSCs, and hES cells were detached/resuspended from the culture flasks with cold PBS and washed again with PBS. Signal acquisition was performed with a FACS Calibur (BD Biosciences), and the cells with FSC/SSC patterns compatible with viable cells were gated. Then, a second selection was done for 7AAD negative population. Analyses were performed using FlowJo software, version 10 (Tree Star). Bovine fibroblasts were trypsinized from 24-well plates, where 200 μl of DMEM was added per well. The contents were transferred to a microtube and centrifuged at $687 \times g$ for 5 min, and the supernatant was removed, leaving ~50 μl for analysis. Protoplasts from each well of 12-well plates were transferred to a microtube and centrifuged at $100 \times g$ for 1 min, and the supernatant was removed, leaving a volume of ~50 μl. The entire contents of each tube were analyzed. Bovine and protoplast cells were analyzed on an imaging flow cytometer (Amnis FlowSight) under 488-nm laser excitation and a power of 60 and 10 mW, respectively. Signal acquisition in bovine cells was determined by channel 3 intensity (filters at 566-635) versus channel 2 intensity (green EGFP reporter emission, 532–555 nm). Signal acquisition in the protoplast population was determined by channel 4 intensity (chloroplast autofluorescence red emission, 610/30 nm) versus channel 2 intensity. Before the first reading, a no-gating population was acquired under intensity channels and the imaging system from the Amnis FlowSight cytometer allowed visually gating the viable cells population. Once the gate was set up, the samples were acquired using this gate. The results from bovine and protoplast cells were processed and visualized using IDEAS software. For mammalian systems, at least 10,000 single cells were analyzed, and for protoplasts, at least 1000 single cells were analyzed.

**DNA extraction, amplification, and sequencing**. For HEK 293T cells, PBMCs, and bovine fibroblasts, total DNA was extracted from the pool of technical triplicates using DNeasy Blood & Tissue Kits (Qiagen) after a 48-h assay. Plant protoplasts were also pooled after a 24-h assay, and the DNA was extracted using the DNeasy Plant Mini Kit (Qiagen). For all samples, the PCRs were carried out using appropriate primer pairs (Supplementary Table 7) to amplify the *attL* and *attR* site-containing regions from the reporter switch vectors (pSG and pSP) using Platinum Taq DNA polymerase (Invitrogen). The Wizard® SV Gel and PCR Clean-Up System (Promega) was used to clean the expected amplicon from the agarose gel. The PCR products were cloned into the pGEM-T Easy vector (Promega) according to the manufacturer's protocol and transformed into *E. coli* DH10B or XL1-blue chemically competent cells by heat shock. The plasmid DNA was extracted by the Wizard® Plus SV Miniprep DNA Purification System (Promega) and sequenced by Macrogen, always using the M13F and M13R or SP6 and T7 primer pair for coverage sequencing. The obtained sequences were aligned with the expected activated plasmid sequences and trimmed using Geneious software (version 7.0.6).

**Cell viability assays**. HEK 293T cells and bovine fibroblasts were plated in 96-well plates at a density of $1 \times 10^5$ cells/well in triplicates and grown for 24 h at 37 °C in a 5% CO₂ atmosphere. The cells were cotransfected with pIE and pSG vectors or only with one of these vectors plus a mock plasmid as previously described proportionally to a 96-well plate assay. For the impairment negative control were used 20 μL dimethyl sulfoxide (DMSO; Sigma-Aldrich) for a final volume of 200 μL per well. After 48 h transient cotransfection, the cells were incubated with 15 μl of 3-(4,5-dimethylthiazol-2-yl)-2,5-diphenyltetrazolium bromide (MTT; Thermo Fisher Scientific) (5 mg/mL) for 4 h at 37 °C. Subsequently, the medium was removed and 150 μl DMSO (Sigma-Aldrich) was added to each well to dissolve formazan crystals. A microplate reader (Sunrise reader, Tecan) calibrated to read absorbance at 595 nm was used to quantify the formazan product. Protoplasts were transformed with pIE vector of each integrase or with a mock plasmid in triplicates as previously described. The impairment negative control was obtained with 50 μL DMSO (Sigma-Aldrich) added per well to complete the final incubation volume of 500 μL. The fluorescein diacetate (FDA; Sigma-Aldrich) assay was performed following Lin et al.[58] protocol with some modifications. After 24 h transient transformation, each sample was incubated with 3 μL FDA work solution [10 μL stock solution (5 mg/mL) in 2.5 mL W1 solution] by 8 min and analyzed by flow cytometry as previously described. The work solution was remade every 2 h. Channel 2 intensity was used to acquire FDA positive cells population in an adequate gate corresponding to viable cells.

**Statistics and reproducibility**. The dataset for the genetically activated cell proportion (EGFP+) obtained by flow cytometry was analyzed using R software (version 3.6.0). The nonparametric Kruskal–Wallis test was used to determine significant differences between controls and test conditions of each Int group at the 5% statistical probability level.

## Data availability

All plasmids constructed for this study are available in Addgene repository and the accession numbers are listed in Supplementary Table 1. Also, the genetic sequence parts used are listed in Supplementary Information. The complete sequence alignment dataset is available in Supplementary Data. The full dataset of positive GFP cell proportion obtained by flow cytometry analysis in all cell experiments and the OD acquisitions of MTT assays were deposited in Dryad Digital Repository (https://doi.org/10.5061/dryad.dr7sqv9tv)[59]. Any other data are available from the corresponding authors upon request.

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

## Acknowledgements

We thank Joseane Padilha da Silva, MSc, from Embrapa Genetic Resources & Biotechnology for providing assistance with the statistical analyses; Dr Miguel Andrade, for figures graphic design; Adriano Caiado A. F. de Almeida, MSc, for contributions to the p35S(rc)2_4_5-GFP (pSP) cloning process; Lara Rodrigues Nesralla, MSc, who took part in the adaptation of the protocols for protoplast extraction and transformation; Dr Andrielle Thainar Mendes Cunha, for assistance with the flow cytometry methodology; and INCA's blood bank and hemotherapy service for the preparation of donor samples. This research was funded by INCT BioSyn (National Institute of Science and Technology in Synthetic Biology), CNPq (National Council for Scientific and Technological Development), CAPES (Coordination for the Improvement of Higher Education Personnel), Brazilian Ministry of Health, and FAPDF (Research Support Foundation of the Federal District), Brazil.

## Author contributions

Conceived and designed the experiments: E.R., C.M.C., M.H.B., and A.M.M. Performed vector design and cloning: C.M.C., M.S.G., G.P.C.J., M.S.M.A., L.M.G.B., and R.N.L. Performed the *Arabidopsis thaliana* protoplast experiments: M.S.G., L.M.G.B., L.H.F., and C.L. Performed the bovine fibroblast experiments: T.T.S., M.A.O., E.O.M., G.P.C.J., and R.N.L. Performed the HEK 293T cell, PBMCs, NSCs, and hES cell experiments: L.R. C.B., C.G.L., M.L.R., L.H.F, M.A.O., and S.K.R. Wrote the paper: E.R., C.M.C., M.S.G., T. T.S., L.M.G.B., R.N.L., and M.H.B. All the authors analyzed, interpreted, and discussed the results and contributed to the final paper.

## Competing interests

The authors declare no competing interests.
