## [Peer Review File · Communications Biology]

Reviewers' comments:

Reviewer #1 (Remarks to the Author):

This paper describes the functionality of six recently described serine integrases in a range of eukaryotic cells. The headline observation is that one of the integrases int13 is as active in vertebrate cells as the currently top performing integrases; those from phages Bxb1 and PhiC31.

Otherwise and in detail:

Line 127; need to emphasize that this is a transient transfection assay and that there will be many copies of both plasmids in the cells, that the ratios will vary and that the approach may be compromised if the integrases are toxic, which in some cases we know them to be.

The sentence beginning on line 165 "For..." is not clear. I think that it means that Int 13 is the most active in HEK293T, perhaps better to just say so.

Line 259; please do not abbreviate PBMC; just write it out in full.

Figures the PCR gel data could be moved to the supplementary data

Figures; the Sanger traces could be moved to supplementary data

Figures; the FACS data could be moved to supplementary data

Overall this is a well written, thorough piece of work that will be of interest to those in the field.

However, the use of transient assays is a limitation that could have been rectified by testing for toxicity, by working with stably transfected cells containing the switch constructs as a single copy and by measuring integrase expression by blotting. These omissions mean that the work described will not be the last word on the characterization of the potential utility of these integrases. In the discussion it would also be of value to describe the Int13 in a bit more detail; which phage does it come from and how does the size of the protein compare with the other integrases?

William Brown

Reviewer #2 (Remarks to the Author):

In this study, the authors generated unidirectional genetic switches to evaluate the functionality of six serine integrases in eukaryotic systems, namely in human embryonic kidney cell lineage (HEK 293T), bovine fibroblasts, and Arabidopsis thaliana protoplasts. The authors also extended their study to test integrase activity in three human peripheral blood mononuclear cells (PBMCs), induced pluripotent stem cell derived-neural stem cells (NSCs) and undifferentiated embryonic stem (ES) cells. Altogether, this article introduces some newly discovered serine integrases (2, 4, 5, 7, 9, and 13) that can be further used for the engineering of the eukaryotic system which can be applied for several biotechnological applications. Overall, the manuscript is well written, and all sections of manuscript are well described. However, the work described in this manuscript is an extension of a previous study (Yang et al 2014), and the novelty of this work is not clearly stated. The following comments should be addressed during revision.

1- Yang et al. have identified 34 putative Ints, demonstrating the functionality for 11 of these Ints in prokaryotic cells. Please provide a rationale behind selecting only 6 specific integrases (Ints 2,4,5,7,9,13) for this study.

2- Yang et al 2014 reported that the Ints 2,4,5,7,13 are completely functional in prokaryotes and yielded 100% expression, shown by the GFP-positive cell populations. However, the current study showed a significant degree of variation in the activity of Ints 2,4,5,7,13 when tested in eukaryotes cell system, also shown by the GFP expression in HEK 293T, bovine fibroblasts, and Arabidopsis

thaliana protoplasts. It will be good if the authors can explain why the activity of selected Ints has a wide range of variation? The authors can also explain how the functionally low active Ints can be helpful in designing genetic networks?

3- In the result section "Promoter as switchable genetic element and orthogonality of the Ints" the authors evaluated a promoter as a switchable part to test the Ints activity in *A. thaliana* protoplast. The authors have excluded Ints 13 in this study although Ints 13 showed a substantial activity. Please provide justification for the selection of Ints and a rationale for using plant for evaluation. As a switchable genetic element (Promoter) is an important part in designing a genetic network in mammalian cells, it is recommended that the activity of Ints be evaluated using mammalian cell lines.

4- The widespread use of six Ints 2, 4, 5, 7, 9, as genetic switches was well investigated and reported by the authors in mammalian and plant cells. Overall this study shows only validation of Ints activity. To justify the overall significance of this work, it may be worthwhile to select one or two active integrases and experimentally determine their application. For example, this can be applied to develop switchable genetic networks in industrially important mammalian cell lines (CHO/ HEK293T) or design a genetic circuit with therapeutic applications in human T lymphocytes or stems cells.

5- Line 333: 'no limiting toxicities were observed in the performed assays.' Please state which assays were performed to evaluate the toxicity and provide supporting evidence.

6- Line 215-217: "Mutations were also observed in the EGFP sequence for most clones sequenced, suggesting that the observed mutations may be due to PCR errors instead of imprecise Int recombination". Please elaborate on the controls used for the PCR reaction or provide experimental basis that supports such a conclusion.

Minor comments:

1- In most of the figures the authors illustrated the chromatogram of sequencing results of the attL and attR sequences obtained after Ints activity. Due to poor resolution of the sequencing chromatogram illustrated in the figures, we recommend to move all this information to the supplementary section of the manuscript. Another suggestion is to show a part of sequencing results of the attL and attR as a normal text form without chromatogram.

2- Line 301- Extra Hyphens should be removed between texts.

3- The manuscript contains a number of typographical and grammatical errors.

Reviewer #3 (Remarks to the Author):

The manuscript by Gomide et al. describes a set of assays to test the site-specific recombination function of six recently identified serine integrases in several different cell types. They also compare the activities of these integrases to the previously well studied phiC31 and Bxb1 integrases. The authors' aim is to validate the usefulness of the new integrases for synthetic biology applications.

The manuscript is well written and clear, and in general the data are clearly presented. As far as I can tell all the experimental work has been carried out carefully and competently, and I am confident that the data/methods used are sound. The results and conclusions will be of some interest to those interested in using serine integrases for various applications in living cells.

Comments

1. Discussion section. Most of the manuscript is very straightforward and clear, but I felt that the Discussion was rather long, and contained too much unnecessary detail. The authors should consider revising it, primarily by making it shorter and focusing more directly in their key 'take-home messages'.

2. Methods section; integrases and plasmids. To make their results useful to the research community, the authors should provide full sequences of the constructs they use (in particular, all the (codon-optimized) integrase open reading frames and the att sites). Although these may be available elsewhere, I think that it would be best for them to accompany the paper as Supplementary Material.

3. Figures - general points. (a) In several of the Figures, text annotation is far too small to read, and will become illegible in standard pdf formats. The authors should ensure that all the text is big enough (for example, on the flow cytometry diagrams and the sequence profiles). (b) Several of the figures show DNA sequencing graphs. These clutter up the figures and seem to be completely unnecessary; I can't see any value of them for the reader as they simply show that the predicted site-specific recombination events have occurred. A properly laid out diagrammatic figure showing each of the pairs of att site sequences and their predicted recombinant products would be much more helpful. (c) In several Figures there are flow cytometry plots, which I found very difficult to understand. I am not an expert in this technique, but I think the authors should give enough explanation so that a general reader can make sense of them. In particular, the units being measured on the y-axis (SSC-H, Ch3, FL3-H, etc) should be defined/explained, and the rationale for defining the 'GFP+' boxes/gates should be explained. (d) In several bar chart figures the bars are annotated with letters (a, b, c, d) which should be properly explained.

4. In Figure 2b and other similar figures, would it be better to express the values as % of the positive control? This might give a more realistic impression of the level of recombination, because as shown the apparent level is greatly reduced presumably due to incomplete transfection of the cells.

5. Supplementary Table 8. The authors' identification of a possible error in the definition of the crossover point in the att sites of Int9 is worth including for the record, but the Table as it is might just confuse many readers. Could the results be shown in a more diagrammatic way that would show the readers clearly how the sites are proposed to be broken and rejoined?

Response to referees letter

Reviewer #1

This paper describes the functionality of six recently described serine integrases in a range of eukaryotic cells. The headline observation is that one of the integrases int13 is as active in vertebrate cells as the currently top performing integrases; those from phages Bxb1 and PhiC31.

Otherwise and in detail:

Line 127; need to emphasize that this is a transient transfection assay and that there will be many copies of both plasmids in the cells, that the ratios will vary and that the approach may be compromised if the integrases are toxic, which in some cases we know them to be.

We agree with the reviewer about the need to emphasize that the experiments were performed as transient assays. The manuscript has been modified accordingly. Modifications in current line 127; 133; 503; 519; 537; 565.

About the toxicity, we also agreed with the reviewer and performed viability tests on HEK293T, bovine fibroblasts and protoplasts. This assay is described below (last part of our answers to the reviewer #1) and the results were included in the manuscript (current lines 189; 217; 612).

The sentence beginning on line 165 “For...” is not clear. I think that it means that Int 13 is the most active in HEK293T, perhaps better to just say so.

The intention in this sentence was to highlight that Int 13 was the integrase that obtained the best performance among the six integrases herein evaluated for the first time in eukaryotic cells (Ints 2, 4, 5, 7, 9 and 13). However the phiC31 and Bxb1 integrases, in HEK 293T, were the ones that promoted the highest number of positive EGFP cells. In fibroblasts, on the other hand, Int 13 exhibited the best performance among all, activating more cells even than phiC31 and Bxb1. The manuscript text was modified seeking a better explanation. Now the modified text states in line 171.

Line 259; please do not abbreviate PBMC; just write it out in full.

We accept the reviewer’s suggestion, and the manuscript was modified accordingly (current line 277).

Figures the PCR gel data could be moved to the supplementary data

Figures; the Sanger traces could be moved to supplementary data

Figures; the FACS data could be moved to supplementary data

We thank the reviewer's proposals. We evaluated the figure changes suggested and, as the transfer of sequence alignments to supplementary information was a common point among all three reviewers' comments, we decided to follow this suggestion, and we agreed that is not so necessary to compound the central figures. However, as transfer the FACS graphics and PCR gels for supplementary information was not a point mentioned by the other two reviewers and, as we agreed that these are an essential set of data for the entire comprehension of the study, we decided to keep these in the central figures. Nevertheless, we did minor alterations to improve the understanding of the data, like increase the font size of the texts associated with the images.

Overall this is a well written, thorough piece of work that will be of interest to those in the field. However, the use of transient assays is a limitation that could have been rectified by testing for toxicity, by working with stably transfected cells containing the switch constructs as a single copy and by measuring integrase expression by blotting. These omissions mean that the work described will not be the last word on the characterization of the potential utility of these integrases. In the discussion it would also be of value to describe the Int13 in a bit more detail; which phage does it come from and how does the size of the protein compare with the other integrases?

Indeed, toxicity is a concern for the use of integrases as a biotechnological tool. Although we could observe by our data that the assays are not compromised, since we had a minimal number of cells measured by flow cytometry in all conditions (controls and test), we agreed with the reviewer that a specific test should be used to measure possible integrase toxicity. Thus, we performed MTT assays for HEK 293T cells and fibroblasts, and FDA assays for protoplasts. The results were added to the manuscript, and they indicated that integrases are not cytotoxic for the model cells evaluated, corroborating with the above cited observation (current lines 189; 217; 612).

Additionally, the main goal of our study was to perform a eukaryotic cell screening of functional integrases to augment the number of those proteins that can be used to activate genetic switches in a variety of eukaryotic systems. We believe that the transient assays are a satisfactory kind of investigation to validate the functionality of these proteins as genetic tools for future uses in synthetic genetic networks or other biotechnological applications. Also, we

consider that stable construction and investigations related this kind of assays can be an approach to compound future works, being out of the scope of the present study.

Regarding the additional information about Int 13, we agree with the reviewer's suggestions that it would be valuable for the readers. The requested information was added to the manuscript discussion (current line 327).

Reviewer #2

In this study, the authors generated unidirectional genetic switches to evaluate the functionality of six serine integrases in eukaryotic systems, namely in human embryonic kidney cell lineage (HEK 293T), bovine fibroblasts, and *Arabidopsis thaliana* protoplasts. The authors also extended their study to test integrase activity in three human peripheral blood mononuclear cells (PBMCs), induced pluripotent stem cell derived-neural stem cells (NSCs) and undifferentiated embryonic stem (ES) cells. Altogether, this article introduces some newly discovered serine integrases (2, 4, 5, 7, 9, and 13) that can be further used for the engineering of the eukaryotic system which can be applied for several biotechnological applications. Overall, the manuscript is well written, and all sections of manuscript are well described. However, the work described in this manuscript is an extension of a previous study (Yang et al 2014), and the novelty of this work is not clearly stated. The following comments should be addressed during revision.

We would like to emphasize that the scientific novelty brought with our study is the validation of functionality of some of these integrases recently identified by Yang et al 2014 and only tested in bacteria. In our research these proteins were shown to be functional in different eukaryotic cells, including those with therapeutic importance, a model for livestock studies, and plant cells, an essential model in a field still taking its first steps towards plants synthetic biology. The leap from bacteria validation to eukaryotic cells validation is a frequent stage to improve genetic tools for different applications.

1- Yang et al. have identified 34 putative Ints, demonstrating the functionality for 11 of these Ints in prokaryotic cells. Please provide a rationale behind selecting only 6 specific integrases (Ints 2,4,5,7,9,13) for this study.

The rationale explaining the six integrases selection was described on line 137 (current line 140). The resulting *attL* sites for the excluded integrases (Ints 3, 8, 11 and 12) had one or more ATG start codon. It would remove the

EGFP frame, or, even with a frame correction, it would lead to a premature translation, and a consequent small peptide added to EGFP, which could compromise the protein functionality. Thus we decided to work just with the six chose integrases, with which this problem would not happen. We also added in the manuscript text information about the exclusion of Int 10, which occurred because of orthogonality errors (current line 142). The only exception for our ATG presence criteria occurred for the switch promoter assay, once Int 2 *attR* site, that in this case is upstream of the *egfp*, presents two ATG in the sequence, but they are in frame with the *egfp* ORF.

2- Yang et al 2014 reported that the Ints 2,4,5,7,13 are completely functional in prokaryotes and yielded 100% expression, shown by the GFP-positive cell populations. However, the current study showed a significant degree of variation in the activity of Ints 2,4,5,7,13 when tested in eukaryotes cell system, also shown by the GFP expression in HEK 293T, bovine fibroblasts, and *Arabidopsis thaliana* protoplasts. It will be good if the authors can explain why the activity of selected Ints has a wide range of variation? The authors can also explain how the functionally low active Ints can be helpful in designing genetic networks?

A comment about the EGFP activation frequencies variation was done on line 316 (current line 337). Besides the mentioned hypothesis of intrinsic complexity of eukaryotic organisms, there are also the technical limitations for cells transformation. Unlike what occurs for bacteria, the transformation efficiencies in eukaryotic cells is variable, as could be noted by frequencies of EGFP expressing cells in positive controls population. Since not all cells are transformed, the result of the transformation is naturally variable. Nevertheless, there could be other reasons for the variable EGFP positive cells frequencies observed, such as NLS absence, zinc chelation by culture medium and consequent interference in zinc ribbon domain of the integrases structure, mRNA degradation, and degradation/interferences in translational or post-translational levels, like other proteins interactions. Also, the complexity of eukaryotic nucleus environment may become difficult the recognition sites meeting and the correct synapse, cleavage and sequence recombination, among other possible questions. The investigation of these possibilities and further basic sciences investigations are essential and can help to improve the biotechnological tool functionality and applicability. However, the number of hypothesis raised means that we do not know if we would be able to answer and how soon we could answer with certainty why the activation rates varied. Even so, with the functionality validation, as done for the serine integrases in eukaryotic cells in the present study, the application is already possible. If the inversion target sequence, for example, is integrated into the genome, the reaction needs to occur once, being necessary only an integrase tetramer.

In this way we can also discuss the possible tunability of the integrase set applications. Some integrases with a lower rate of switch activation may be of interest in synthetic genetic networks when such a rapid activation is not desired. It may be necessary to delay the activation of one piece concerning another. For example, once there is a necessary cellular growth first and later the production of the desired protein. If the integrase function slowly it would give time for cell proliferation before the target gene is activated to produce the protein of interest. It would be especially helpful if the product impairs or competes with cell growth.

3- In the result section “Promoter as switchable genetic element and orthogonality of the Ints” the authors evaluated a promoter as a switchable part to test the Ints activity in *A. thaliana* protoplast. The authors have excluded Ints 13 in this study although Ints 13 showed a substantial activity. Please provide justification for the selection of Ints and a rationale for using plant for evaluation. As a switchable genetic element (Promoter) is an important part in designing a genetic network in mammalian cells, it is recommended that the activity of Ints be evaluated using mammalian cell lines.

First of all, we decided to make this construction with three integrase recognition sites in tandem flanking another genetic piece (promoter) for additional answers regarding orthogonality and no system commitments with more than one recognition site presence. Thus, we decided that three integrases would be an adequate number for these further analyses. We did not choose Int 13 for two reasons: i) there is an ATG start codon in the Int 13 *attR* site formed after recombination, that would be positioned upstream the *egfp* ORF, removing the gene sequence frame; ii) Yang et al 2014 reported a constitutive promoter activity for Int 13 *attP* site. They solved the problem swapping *attB* and *attP* sites positions. However, as we would work with a part that already had a risk of leak activity in an inverted position (promoter), we decided not to take the risk of a double leak. Considering that, we chose to design the pSP vector with another integrase that also showed useful functionality (Int 4), with one with weak feature (Int 2), and with Int 5 whose functionality was previously observed only by molecular analyses. We questioned ourselves if it would also happen with another part inversion. We modified the text to give a better explanation why Int 13 was not chosen, as follows in current line 247.

Second, we mentioned in line 231 (current line 244) that we chose plant model because it was the system with higher number of cells expressing EGFP, indicating robustness in Int activity. It was our first decision, but we agreed with the reviewer that analyzing promoter inversion is an essential approach for the mammalian systems as well. We also had built a pSP vector to the mammalian

system, with the EF1alpha promoter on a reverse complement position flanked by the same Int sites of plant construction in tandem. However, in the mammalian system a high number of cells EGFP positive was observed just with the pSP plasmid (negative control), indicating a higher leak level for this promoter than the low one observed for the plant system. As for solving this situation, we would work with promoter modifications, and as it was not the primary goal of our study, we decided not to advance with the inverted promoter system evaluation in the mammalian cells.

4- The widespread use of six Ints 2, 4, 5, 7, 9, as genetic switches was well investigated and reported by the authors in mammalian and plant cells. Overall this study shows only validation of Ints activity. To justify the overall significance of this work, it may be worthwhile to select one or two active integrases and experimentally determine their application. For example, this can be applied to develop switchable genetic networks in industrially important mammalian cell lines (CHO/ HEK293T) or design a genetic circuit with therapeutic applications in human T lymphocytes or stems cells.

We agree that we gave the first and essential step toward the functional validation of these integrases to be used as genetic tools in different and relevant eukaryotic models. To build genetic networks for industrial and therapeutic applications is the second step that can be developed for other research groups using the information brought for our study, and also encompasses the next steps of research in our group, that is a focus for future publications.

5- Line 333: 'no limiting toxicities were observed in the performed assays.' Please state which assays were performed to evaluate the toxicity and provide supporting evidence.

We described no limiting toxicities caused by the integrases supported by our microscopy observations and by no significant differences observed on cell quantifications among all conditions (controls and test groups). Nevertheless, we agreed with the reviewer #2 and also with the reviewer #1 that a specific viability test would better support this information. Thus we performed MTT assays for HEK 293T cells and fibroblasts, and FDA assay for protoplasts. The results were added to the manuscript (current lines 189; 217; 612).

6- Line 215-217: "Mutations were also observed in the EGFP sequence for most clones sequenced, suggesting that the observed mutations may be due to PCR errors instead of imprecise Int recombination". Please elaborate on the controls

used for the PCR reaction or provide experimental basis that supports such a conclusion.

The PCR strategy was composed by primers that only amplified the inverted genetic part (*egfp* or promoter). Because of that, the only amplicons obtained were from test conditions. Our conclusions about *egfp* mutations were supported by the experimental data provided by the sequencing alignment and by the quantified covered mutations. We reached that proposal because the *egfp* sequence size is some times higher than the sites sequence, increasing the possibilities of occasional PCR errors; also, because the SNPs were in different and random positions, and because the chances of mistakes are higher to happen at the recognition site than inside the genetic part flanked by the sites.

Minor comments:

1- In most of the figures the authors illustrated the chromatogram of sequencing results of the attL and attR sequences obtained after Ints activity. Due to poor resolution of the sequencing chromatogram illustrated in the figures, we recommend to move all this information to the supplementary section of the manuscript. Another suggestion is to show a part of sequencing results of the attL and attR as a normal text form without chromatogram.

We agreed with the reviewer suggestions, and the chromatogram sequencing figures were transferred for supplementary information. Also, they were modified for a better visualization and comprehension (current Supplementary Figures 5, 7, 9 and 14).

2- Line 301- Extra Hyphens should be removed between texts.

The dash used to separate the named integrases were a correction proposed by springer nature English language editing service. However, as this typography was not so suitable, we changed as proposed by the reviewer (current line 320).

3- The manuscript contains a number of typographical and grammatical errors

As previously mentioned, we used the springer nature English language editing service. However, for improve our text comprehension we sought to correct the still present grammatical and typographical errors in the manuscript.

Reviewer #3

The manuscript by Gomide et al. describes a set of assays to test the site-specific recombination function of six recently identified serine integrases in several different cell types. They also compare the activities of these integrases to the previously well studied phiC31 and Bxb1 integrases. The authors' aim is to validate the usefulness of the new integrases for synthetic biology applications.

The manuscript is well written and clear, and in general the data are clearly presented. As far as I can tell all the experimental work has been carried out carefully and competently, and I am confident that the data/methods used are sound. The results and conclusions will be of some interest to those interested in using serine integrases for various applications in living cells.

Comments

1. Discussion section. Most of the manuscript is very straightforward and clear, but I felt that the Discussion was rather long, and contained too much unnecessary detail. The authors should consider revising it, primarily by making it shorter and focusing more directly in their key 'take-home messages'.

We thank the reviewer's suggestion. To this end we sought to do some changes to make the discussion more concise. However, no significant structural modifications were performed to avoid compromising the communication and discussion we proposed ourselves to do. Also, we had to include additional information required.

2. Methods section; integrases and plasmids. To make their results useful to the research community, the authors should provide full sequences of the constructs they use (in particular, all the (codon-optimized) integrase open reading frames and the att sites). Although these may be available elsewhere, I think that it would be best for them to accompany the paper as Supplementary Material.

We agreed with the reviewer and added the genetic sequences used to build the vectors to the supplementary information.

3. Figures - general points. (a) In several of the Figures, text annotation is far too small to read, and will become illegible in standard pdf formats. The authors should ensure that all the text is big enough (for example, on the flow cytometry diagrams and the sequence profiles). (b) Several of the figures show DNA sequencing graphs. These clutter up the figures and seem to be completely unnecessary; I can't see any value of them for the reader as they simply show that the predicted site-specific recombination events have occurred. A properly laid out diagrammatic figure showing each of the pairs of att site sequences and their predicted recombinant products would be much more helpful. (c) In several Figures there are flow cytometry plots, which I found very difficult to understand. I am not an expert in this technique, but I think the authors should give enough explanation so that a general reader can make sense of them. In particular, the units being measured on the y-axis (SSC-H, Ch3, FL3-H, etc) should be defined/explained, and the rationale for defining the 'GFP+' boxes/gates should be explained. (d) In several bar chart figures the bars are annotated with letters (a, b, c, d) which should be properly explained.

a) We increased the font size of the figure associated texts.

b) The importance of the sequencing figures was to show the correct formation of the recombined *attL* and *attR* sites, indicating accuracy of the integrases recombination process. However, we agreed with the reviewer that would be better move these figures to supplementary information. We also highlighted the predicted sequence and the representative one obtained after sequencing alignment, removing the chromatogram peaks for a clear visualization (current Supplementary Figures 5, 7, 9 and 14).

c) We provided additional information about flow cytometry in Methods section (current lines 572 and 587).

d) We explained the letters meaning in the figure legends. The different letters indicate significant statistical differences inside each integrase group.

4. In Figure 2b and other similar figures, would it be better to express the values as % of the positive control? This might give a more realistic impression of the level of recombination, because as shown the apparent level is greatly reduced presumably due to incomplete transfection of the cells.

As the raw data is itself expressed as relative numbers, we preferred do not perform their normalization in order to run the statistical analysis more properly.

5. Supplementary Table 8. The authors' identification of a possible error in the definition of the crossover point in the att sites of Int9 is worth including for the record, but the Table as it is might just confuse many readers. Could the results be shown in a more diagrammatic way that would show the readers clearly how the sites are proposed to be broken and rejoined?

We tried to improve the comprehension about the Int 9 core modification found by our sequencing data with a new diagram showed in the added Supplementary Figure 15.

REVIEWERS' COMMENTS:

Reviewer #2 (Remarks to the Author):

The authors have addressed all the major and minor comments and have revised their paper accordingly.

One suggestion is as follows. Serine integrases (2, 4, 5, 7, 9, and 13) discovered in this study can be employed to engineer eukaryotic systems, and can lead to several applications in the area of synthetic biology. Please provide further discussion on the application of serine integrases in the context of mammalian and plant research in the discussion section.

Response to referees letter

Reviewer #2 (Remarks to the Author):

The authors have addressed all the major and minor comments and have revised their paper accordingly.

One suggestion is as follows. Serine integrases (2, 4, 5, 7, 9, and 13) discovered in this study can be employed to engineer eukaryotic systems and can lead to several applications in the area of synthetic biology. Please provide further discussion on the use of serine integrases in the context of mammalian and plant research in the discussion section.

We would like to thank reviewer #2 remarks. We discussed and provided more specific examples of serine integrase applications as required. Please find the changes in lines 378 and 400.